**Comparison of debris-flow observations, including fine sediment grain size and**
**composition, and runout model results at the Illgraben, Swiss Alps**

[1,2]Daniel Bolliger, [1]Fritz Schlunegger[*], [3]Brian W. McArdell
[1]Institute of Geological Sciences, University of Bern, Switzerland
[2]Geotest AG, Zollikofen, Switzerland
[3]Swiss Federal Institute WSL, Switzerland
*Corresponding author: fritz.schlunegger@unibe.ch
**Abstract**
Debris flows are important processes for the assessment of natural hazards due to their
damage potential. To assess the impact of a potential debris flow, parameters such as the
flow velocity, flow depth, maximum discharge and the volume are of great importance. This
study uses data from the Illgraben observation station, Central Alps of Switzerland, to explore
the relationships between these flow parameters and the debris flow dynamics. To this end,
we simulated previous debris flow events with the RAMMS debris flow runout model, which is
based on a numerical solution of the shallow water equations for granular flows using the
Voellmy friction relation. Here, the events were modeled in an effort to explore possible
controls on the friction parameters $\mu$ and $\xi$, which describe the Coulomb friction and the
turbulent friction, respectively, in the model. Additionally, sediment samples from levee
deposits were analyzed for their grain size distributions (14 events) and their mineralogical
properties (four events) to explore if the properties of the fine-grained matrix have an influence
on the debris flow dynamics. Finally, field data from various debris flows such as the flow
velocities and depths were statistically compared with the grain size distributions, the
mineralogical properties, and the simulation results to identify the key variables controlling the
kinematics of these flows. The simulation results point to several ideal solutions, which depend
on the Coulomb and turbulent friction parameters ($\mu$ and $\xi$, respectively). In addition, the
modelling results show that the Coulomb and turbulent frictions of a flow are related to the
Froude number if the flow velocity is < 6-7 m/s. It is also shown that the fine-sediment grain
size or clay-particle mineralogy of a flow neither correlates with the flow's velocity and depth,
nor can it be used to quantify the friction in the Voellmy friction relation. This suggests that the
frictional behavior of a flow may be controlled by other properties such as the friction generated
by the partially fluidized coarse granular sediment. Yet, the flow properties are well-correlated
with the flow volume, from which most other parameters can be derived, consistent with
common engineering practice.

## 1 Introduction

### 1.1 Debris flows, and parameters controlling their velocity and runout

Debris flows are rapid mass movements consisting of water-saturated and poorly sorted debris with a large range of grain sizes. Debris flows tend to develop one single or a suite of multiple surges with steep coarse-grained fronts. Their motion is driven by gravity and resisted by friction within the flow and at the boundary with the channel bed (Iverson, 1997). The boulder-rich front is then followed by a tapering body where the pore fluid pressures are large, often exceeding the hydrostatic pressure (Iverson, 1997, McArdell et al., 2007). In the frontal part, larger particles tend to ascend in the debris flow body due to particle collisional stresses thereby building a coarse-grained top layer (Johnson et al., 2012), which travels somewhat faster than the flow front itself, delivering coarse sediment to the front. Accordingly, the coarser-grained particles along with some of the fine sediment present at the surface of the flow tend to accumulate in the surge head and are deposited laterally in levees just a few meters behind the front (Johnson et al., 2012).

In the past years, de Haas et al. (2015) conducted experiments to investigate how the grain size distribution and water content influences the velocity of a debris flow. They found that a higher clay content tends to result in an increase of both the velocity and the runout distance of such flows. However, if the clay content becomes too large, then the velocity decreases due to a higher viscosity of the fluid. This relationship should also be applicable to the silt fraction because clay and silt particles are a part of the fluid while grains larger than silt contribute to the solids of a debris flow (Iverson, 1997). The experiments of de Haas et al. (2015) also showed that a large gravel content in the flow front leads to a strong frictional resistance, which in turn reduces the flow velocity. In addition, a large gravel content results in a larger pore water diffusivity, which reduces the pore pressure in the flow and contributes to a further reduction of the flow velocity. On the other hand, a low gravel content leads to lower collisional forces, which might also lead to a relatively low flow velocity. Furthermore, also according to the experiments by de Haas et al. (2015), the water content, the velocity, the volume, and the runout distance of a debris flow are positively correlated to each other. Based on a combination of experimental and field data, Hürlimann et al. (2015) came to the same conclusions, and they additionally found that an increase in the clay content generally leads to a reduction in the runout distance. Indeed, the absorption of water in swelling clay minerals has the potential to result in an increase of the cohesion of a flow, which in turn could cause a reduction of the flow velocity and the runout distance. Finally, using laboratory experiments, Kaitna et al. (2016) documented that a relatively high fraction of fine-grained material tends to occur in flows with excess pore fluid pressures. In addition, these authors mentioned that such flows were characterized by low fluctuations of normalized fluid pressures and normal stresses, and the experiments showed that the shear stresses were concentrated

at the base of the flow. Based on the conclusions of the aforementioned authors, we expect
to see a dependency of the flow properties in the Illgraben and the granulometric composition
of these flows (Uchida et al., 2012), and we anticipate that the flow velocity is negatively
correlated with the relative abundance of the finest-grained particles.
The mineralogical composition of a flow is a further parameter, which has the potential to
impact the rheology and thus the flow velocity and runout distance of debris flows, yet these
relationships have largely been overlooked in the literature. In particular, because clay
minerals are important constituents of the fine-grained fraction of these flows, they have the
potential to regulate the pore fluid pressure and the stress state through their ability to absorb
water in their crystal structure (Di Maio et al., 2004). This is mainly the case for swelling clay
minerals (see also section above) such as those of the smectite group (Di Maio et al., 2004),
where the pore fluid composition has a large influence on the volume and the shear strength
of these minerals (Chatterji and Morgestern, 1990; Di Maio, 1996). Because shear stresses
within a flow are a direct consequence of the friction between the particles and the fluid phase
and since the friction properties directly influence the propagation of a debris flow (see section
1.2), we anticipate the occurrence of a direct relationship between the velocity and runout
distance of debris flows, and the mineralogical composition of the fine-grained matrix.

*1.2  Physically-based models describing debris flow processes, and goal of paper*
There are several rheological models or flow resistance relationships describing the behavior
of debris flows such as the flows' velocities, runout distances and frictional properties (e.g.,
Allen, 1997; Rickenmann, 1999; Naef et al., 2006). One commonly used approach is the
Voellmy friction relation (Voellmy, 1955; Salm, 1990; 1993; Christen et al., 2012), which is
also implemented in the software RAMMS, a software package to simulate debris flow runout
(see section 3.1). In the Voellmy friction equation, the frictional resistance of a flow $S$ [Pa] is
composed of the sum of two friction terms: (i) A dry Coulomb-type friction term, referred to as
Coulomb friction, describes the frictional resistance between the debris flow and the channel
bed and mainly depends on the flow depth; and (ii) a drag or viscous-turbulent friction term
describes the turbulent frictional resistance, which mainly depends on the dynamic pressure
and thus on the velocity of the flow. Both components are characterized by the coefficients $\mu$
and $\xi$, which control the values of the Coulomb and the turbulent frictions, respectively
(Christen et al., 2012). Optionally, cohesion stresses can be included in an extended Voellmy
friction equation (Bartelt et al., 2015; Berger et al., 2016). Because this additional cohesion
term has rarely been used in engineering practice and is apparently relatively small (Berger
et al., 2016), it was neglected herein, and the friction equation takes the following form:
$$S = \mu N + \frac{\rho g v^2}{\zeta}, N = \rho g h \cdot \cos(\varphi) \qquad\qquad (1),$$
where $S$ is the frictional resistance [Pa], $\rho$ the density of the debris flow, $h$ the flow height (or
flow depth), $g$ the gravitational acceleration, $\varphi$ the slope angle of the channel bed, and $v$ the
velocity of the flow.
A simplified approach to characterize a debris flow is the Froude number, which describes the
ratio between the inertial and the gravitational forces:
$Fr = \frac{v}{\sqrt{gh}}$ (2),
where $Fr$ is the Froude number, $v$ the velocity of the flow, $g$ the gravitational acceleration and
$h$ the flow height (Hübl et al., 2009; Choi et al., 2015).
As mentioned above, the velocity and runout distance of debris flows are likely to depend on
the frictional resistance in such mass movements. This friction, in turn, can be characterized
by two coefficients $\mu$ and $\xi$ in the Voellmy friction relation (1). Because we anticipate that the
mineralogical and granulometric composition of the fine-grained matrix has an influence on
the properties of such flows (see section 1.1), we expect to identify a relationship between the
frictional properties of a flow, its velocity, and its grain size and mineralogical composition.
Here, we test and explore these hypotheses using in-situ data collected at the Illgraben debris
flow monitoring station situated in the Central European Alps (Figure 1), and we evaluate the
data with the results of a numerical runout model referred to as RAMMS. Upon combining field
data with modelling results, we aim at identifying those parameters that have the largest
control on the dynamic properties of the debris flows at the Illgraben.

**2   Study site and setting**
The Illgraben catchment is located in the Valais region in western Switzerland (Figure 1). It
extends from the summit of the Illhorn (2716 m asl) to the outlet of the Illgraben into the Rhone
River (610 m asl). The total area of about 9.5 km$^2$ consists of the Illgraben basin, which has a
spatial extent of 4.6 km$^2$, and the Illbach tributary catchment covering 4.9 km$^2$ (Figure 1a). The
Illgraben basin has been very active and has generated several debris flows each year
(Schlunegger et al., 2009; McArdell and Satori, 2022). The rates of sediment discharge in the
Illgraben have been exceptionally high for Alpine standards (Berger et al., 2011a). Several
studies showed that the erosion rates and the numbers and extents of debris flows strongly
depend on the hydro-climatic parameters such as the average annual temperature and the
precipitation rates (Bennett et al., 2013; Hirschberg et al., 2019; 2021a, b).The highly fractured
bedrock (Bumann, 2022), belonging to the Penninic nappe stack (Gabus et al., 2008), consists
of massive-bedded limestones, quartzites and Triassic schists with dolobreccia interbeds.
Schlunegger et al. (2009) considered these lithologies to be the main source of the silt and
clay fraction that constitute the matrix of the debris flow deposits. Based on a petrographic
analysis of the debris flow deposits, these authors also identified two distinct sediment sources
in the Illgraben where bedrock lithologies with different petrological properties are exposed.
These are (i) a heavily fractured and foliated suite of gneisses and schists, which are exposed
on the southern flank of the Illgraben, and (ii) a vertically plunging succession of limestones,
dolomites and cellular dolomites, which make up the northwestern flank of the Illgraben
(Figure 1). The material from these two sources is very well mixed in response to repeated
deposition and remobilization of sediment within the catchment (Schlunegger et al., 2009).
The sediment cascade has been subject to seasonal variations, where smaller debris flows
events are associated with net sediment accumulation in the channel, while large flows can
entrain sediment up to several times their initial mass along their flow paths (Berger et al.,
2010; Berger et al., 2011a, b; Schürch et al., 2011).
Grain size analyses conducted on samples from the channel bed and debris flow deposits
indicated sand contents of 35-40% and clay contents of < 5% (Hürlimann et al., 2003;
Schlunegger et al., 2009; Uchida et al., 2021). In the Rhone valley, an alluvial fan has formed
covering an area of 6.6 km$^2$ (Schürch et al., 2016). The channel on the fan has a U-shaped
cross-sectional geometry with a base that is about 5–10 m wide. For the lowermost 2 km, the
gradient of the Illgraben channel ranges from about 7% to 18% (measured over a length of 50
m) with a mean of about 8% (Schlunegger et al., 2009). Thirty-one check dams with vertical
drops of up to several meters were constructed along the lowermost 4.8 km of the channel to
prevent the flows to further incise into the substratum (McArdell et al., 2007; Badoux et al.,
2009). A debris flow monitoring station, situated on the lower fan c. 200 m upstream of the
confluence with the Rhone River, was installed in 2000 and has been operated by the WSL
since then (Hürlimann et al., 2003; Badoux et al., 2009). At the survey site (Figure 1c), the
measured parameters include frontal velocity, flow depth, bulk density, maximum discharge
rate, volumes, and normal and shear force (McArdell et al., 2007; McArdell 2016). The related
values are presented in the openly accessible database of the WSL (McArdell et al., 2023),
and the data was collected using the methods presented in section 3.1. On average 3 to 5
debris flows have been registered by the measuring station every year. They have generally
occurred during intense rainstorms between May and October (e.g., McArdell et al., 2007).

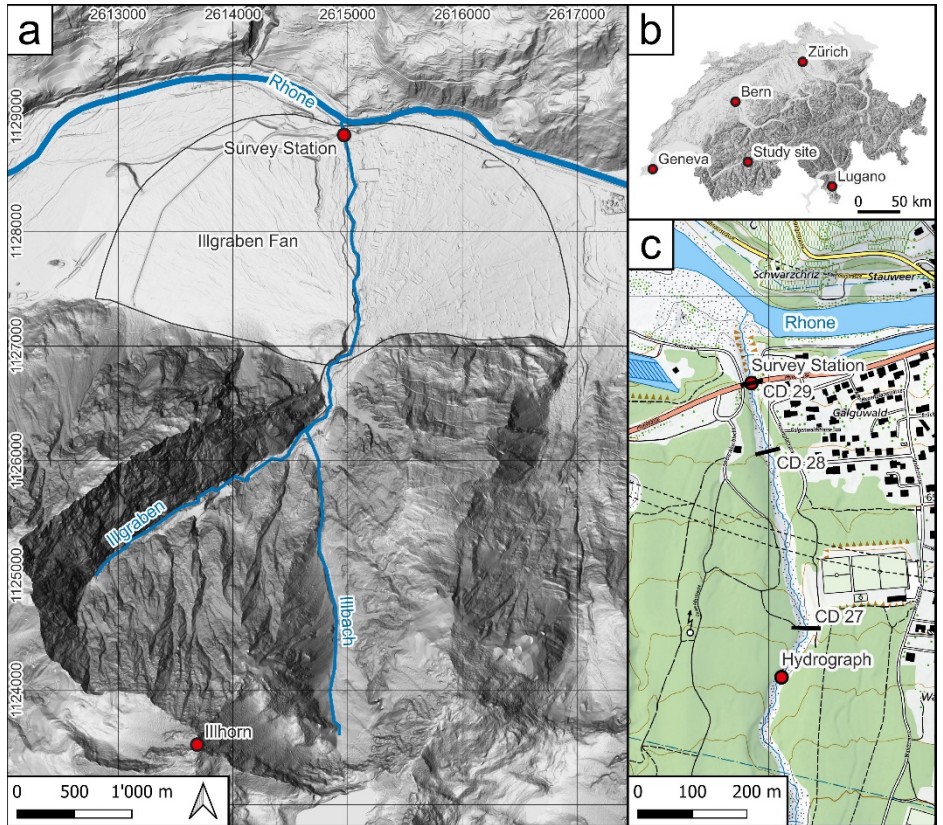

Figure 1: (a) Overview of the topographic situation around the Illgraben showing the Illgraben system with its river network consisting of the Illgraben, Illbach and the Rhone River. (b) Overview map of Switzerland with location of the study site. (c) Detailed topographic map of the Illgraben reach along which the RAMMS simulations have been conducted. It shows the location of the input hydrograph which was used as starting position for the modelling. The three check dams (CD) and the location of the survey station below the Pfynstrasse are displayed on this figure. The background is provided by the swissALTI3D and the Swiss Map Raster 10 (Swisstopo, 2022).

## 3    Methods

Using data collected in the field (section 3.1), we explored how the frictional properties of a debris flow influences the behavior (flow depth and velocity) of such a flow through modelling with RAMMS (section 3.2). We then tested whether the grain size distribution (section 3.3) and the mineralogical composition of the debris flow material (section 3.4) have an influence on the flow velocity.

*3.1    Surveys of debris flows*

Many of the in-situ measurements of the debris flow properties at the Illgraben have been accomplished with a force plate that is installed in the channel beneath a bridge c. 200 m upstream of the confluence with the Rhone River (survey station, see Figure 1c). At that survey site, information on (i) the velocity, (ii) the flow depth, (ii) the mean bulk density, (iii) the duration of individual debris flows, and (iv) the volumes of each flow have been determined in the past years by the WSL (e.g., McArdell and Sartori, 2021; de Haas et al., 2022, Belli et al., 2022; McArdell et al., 2023). As outlined in McArdell et al. (2007) and Schlunegger et al. (2009), the force plate is a horizontal 8 m$^2$ steel structure, which is installed flush with the river

bed just on the top of the concrete check dam. The plate is equipped with normal and shear force transducers. The flow depth is estimated using either a laser or radar unit. Because the radar data is biased by an unpredictable smoothing of the flow surface, we preferentially used the laser data for further calculations. Based on information about the flow depth and the normal force, it was possible to determine the bulk density of a flow as it moves on the plate itself. The volume of each flow was then calculated as the product between the velocity and the cross-sectional area, and this product was integrated over the flow's duration (McArdell et al., 2023). The frontal velocity is determined using the travel time of the flow front over the reach upstream of the force plate (between check dams 27 or 28 and 29; Figure 1c and Hürlimann et al., 2003). Appendix A presents a list of parameters, which been measured at the Illgraben monitoring site, and Appendix B for screenshots from video recordings of selected debris flows.

### 3.2  Numerical modelling with RAMMS

We explored, through modelling with RAMMS, how the frictional properties of a debris flow influence its behavior such as flow depth and velocity. The RAMMS model was developed by the Swiss Federal Institute for Forest, Snow and Landscape Research, WSL (WSL, 2022). It is based on the two-parameter Voellmy-fluid model (Christen et al., 2012; Bartelt et al. 2015), which describes the friction in the 2D depth-averaged equations of motion, which were deviated for granular flows. We justify the selection of such an approach because in an independent modelling study (FLATModel) calibrated with field data (Medina et al., 2007), the Voellmy-fluid formula (eq. *1*) has been proven to reproduce the dynamics of debris flows (flow velocity, erosion pattern in the channel, and aerial extension of the flow in the accumulation zone) reasonably well. A major challenge for modelling is the choice of the input friction coefficients. In particular, if the simulation cannot be calibrated with data that were collected from a previous well-documented event (Christen et al., 2012; Deubelbeiss & Graf, 2011), the input parameters have to be estimated. Because the model results such as the velocity, the runout distance, and the flow depth are sensitive to the friction parameters $\mu$ and $\xi$ (Bartelt et al., 2015; Christen et al., 2012), we iteratively changed the values of these coefficients until we found, for each event, a best fit between the simulation results and the observations (Appendix C).

The Coulomb friction coefficient $\mu$ is sometimes expressed as the tangent of the internal shear angle (WSL, 2022). According to Salm (1993) an internal movement parallel to the slope is only possible if the internal shear angle is smaller than the slope angle. Consequently, the value of $\mu$ should be smaller than the tangent of the channel slope angle. For a minimum slope angle of 7% (4°), $\mu$ should thus be smaller than 0.07. Therefore, for every debris flow event, we conducted several simulations (between 12 and 43, Appendix C) with $\mu$ varying from 0.01

to 0.06, and we modified the $\xi$ parameter to minimize the z-value, which is explained with eq.
(*3*) below. This resulted in $\mu$–$\xi$ pairs with lowest z-values and thus best fits between the model
results and observations (such as the flow velocity *v* and the flow depth *h*). Please see
Appendix C for information about the number of modelling runs, the intervals between the $\mu$–
and $\xi$–values, and other input parameters that we used upon modelling.
We employed the 'hydrograph' input option of RAMMS to characterize the debris flows in the
model. Upon modelling, the hydrograph input was placed c. 500 m upstream of the survey site
(check dam 27, Figure 1c). Erosion was allowed to occur along the entire channel (Frank et
al., 2015; 2017) except at the check dams. Similar to the surveys in the field (section 3.1), the
model velocity was calculated using the travel time between check dams 28 and 29 (Figure
1c). The modelled flow depth values used herein were obtained as the average of the
measurements that were conducted at four points along a cross section at check dam 29.
The use of RAMMS requires a digital elevation model (DEM), event volume, and peak
discharge. The drone-based DEM, which is based on a survey conducted on the 10[th] of August
2021 (de Haas et al., 2022), was used for all simulations. However, this high-resolution DEM
did not cover the section of the channel between the Survey Station and the Rhone River
(Figure 1c). Therefore, in order to extend the area towards the confluence with the Rhone
River, the photogrammetry DEM, which has resolution of 0.1 m, was combined with an existing
0.5 m-lidar DEM (Swisstopo, 2022) using the software QGIS. Here, we resampled the drone-
based DEM to achieve the same resolution as the lidar DEM of Swisstopo (i.e., 0.5 m) so that
both datasets could be combined. The DEM of the short, concrete channel section beneath
the road bridge had to be reconstructed manually because it was not possible to image the
topography below the bridge. In addition, a filter (Serval-Raster editing tools, version 3.10.2)
was applied to the channel bed to smoothen the bed surface. This was done because a large
local change in the topography (such as a boulder) can induce strong vertical accelerations in
RAMMS (and other models that are based on the depth-averaged equations of motion), which
can lead to unrealistically large (or small) local flow depths.
Finally, we introduced a dimensionless *z*-value to describe the deviation of the simulated
velocity *v* and flow depth *h* from the measurements in the field:
$$z = \sqrt{\left(\frac{v_{simulation} - v_{measured}}{v_{measured}}\right)^2 + \left(\frac{h_{simulation} - h_{measured}}{h_{measured}}\right)^2} \qquad\qquad (3).$$
We thus explored how the model input parameters ($\mu$– and $\xi$–values) affect the modelled
velocity and depth values of a flow. We then compared the model results with the surveyed
velocities and depths of each flow using eq. (*3*), which we implemented in the software Matlab
(R2021b).

## 3.3  Grain size distribution

For most of the debris flows that occurred in the years 2019, 2021 and 2022, at least one sediment sample of 1.5 to 3 kg was taken from the levee deposits at the same site labelled as 'Survey Station' in Figure 1c (Swiss coordinates: 2'614'973, 1'128'842; Figure 1c). We collected the material from underneath the bridge to prevent effects related to grain-size-dependent erosion by rainfall. We selected the levee deposits for three reasons. First, according to our experience, the levee deposits can better be attributed to a specific event than other sediments of a debris flow. Second, the levee deposits are those sediments of a debris flow that most clearly record the granulometric composition of the surge head, as our observations on video recordings have shown. Third, it is the surge head, which exerts the greatest control on the dynamics of a debris flow (McArdell et al., 2007; Johnson et al., 2012). Accordingly, upon collecting material from levee deposits, we are likely to analyze sediments with the highest potential to provide information that allows us to understand the dynamics (e.g., flow depth and velocity) of past debris flows. Yet we acknowledge that this material is more likely coarser grained than the sediments in the tail of such a flow (McArdell et al., 2007). In the laboratory, all of the collected material was processed following the state-of-the-art protocol (SN670 004–2b–NA norm), which was established at the Bern University of Applied Sciences (Burgdorf). Following this protocol, the material was first dried and then sieved to a minimum particle size of 0.5 mm using a set of 7 sieves, each of which has a defined mesh size: 31.5 mm, 16 mm, 8 mm, 4 mm, 2 mm, 1 mm, and 0.5 mm. Subsequently, a slurry analysis was carried out on the material < 0.5 mm using a hydrometer. The goal of this task was to determine the particle size distribution between 0.1 and 0.001 mm. Finally, the grain size distribution of the remining material between 0.5 and 0.063 mm was determined by wet sieving. During this task, we used three sieves where the mesh size was 0.25 mm, 0.125 mm and 0.063 mm. The grain size distribution was truncated at 16 mm so that the entire sample is at least 100 times the mass of the largest particle (e.g., Church et al., 1987). We note, however, that particles larger than 16 mm do occur on the levee deposits and we did sample such material in the field. However, we were not able to consider this fraction due to technical limitations in our laboratory and practical limitations on the mass of the sample necessary for analysis.

## 3.4  Powder XRD

We hypothesize that clay minerals influence the pore pressure of a flow (Barshad, 1952), which in turn could influence its mobility (McArdell et al., 2007). We expect such a control because swelling clays tend to absorb water in their crystal structure. The result is an increase in the viscosity of the flow, thereby reducing the dissipation of the fluid pore pressure. To test this hypothesis, the mineralogical properties of some debris flow samples were measured

through standard powder XRD at the Institute of Geological Sciences of the University of Bern. For this purpose, four samples were chosen from fast and slow velocity flows as well as from deposits where either the coarse-grained or the fine-grained fractions dominate in the analyzed grain size spectrum. To this end, the grain size fraction < 0.063 mm, which was already extracted during the steps outlined above, was analyzed for powder XRD. Subsequent milling with a vibrating-disc mill (Retsch RS 200) and a McCrone XRD-mill reduced the particle sizes to the sub-micrometer scale. Corundum powder was added as standard to the samples, and the samples were measured with the x-ray diffractometer X'Pert Pro MPD with Cu radiation. Because this step did not include a determination of the mineralogic composition of the clay minerals, a slightly different approach had to be employed. Here, we used the same initial material, but it was only milled with the vibrating disc mill. The powder was then mixed with a dispersant (0.1 molar $NH_3$) to achieve a homogeneous suspension. The clay particles were separated in an Atterberg cylinder. The particles still in suspension after 15 hours were extracted using a centrifuge. The extracted clay particles were then cleaned with HCl, $CaCl_2$ and deionized water. To distinguish between the different clay minerals, three sample holders were either air dried, treated with ethylene glycol or heated to 400°C and 550°C before measuring with the X'Pert Pro MPD with Cu radiation. The final processing of the data was carried out with the software TOPAS (Coelho, 2018), which uses a Rietveld structure refinement technique (Rietveld, 1969).

## 4    Results

### 4.1    Survey results

A total of 13 events from 2019, 2021, and 2022 were analyzed (Appendix A, Table 1). The measured flow velocities varied by one order of magnitude from 0.89 m/s to 8.69 m/s. The maximum flow depths ranged from 1.13 m to 3.13 m, and the Froude numbers spanned the interval between 0.27 to 2.35, pointing towards considerable differences in the dynamics of these flows. The total volumes reached a maximum of c. 176,000 $m^3$ and the maximum discharge rate was c. 190 $m^3$/s. The measured density ranged from 1189 kg/$m^3$ to 2323 kg/$m^3$, and the corresponding volumetric water contents were between c. 20% and 90%.

Table 1: Measured and analyzed debris flow events from 2019, 2021, and 2022. Velocity, flow depth, volume, maximum discharge (Qmax) and density are the results of direct measurements at the monitoring station in the Illgraben (Figure 1c). The Froude number was derived from these. The last two columns show, for which events XRD analyses and RAMMS simulations were performed. The event of the 26th of July 2019 could not be simulated due to the high Froude number.

| Event date | Velocity [m/s] | Flow depth [m] | Froude number [ ] | Volume [$m^3$] | Qmax [$m^3$/s] | Density [kg/$m^3$] | XRD analysis | RAMMS simulation |
|---|---|---|---|---|---|---|---|---|
| 21.06.2019 | 6.62 | 3.13 | 1.19 | 97394 | 147.61 | 1870 | | ✓ |
| 02.07.2019 | 3.86 | 1.75 | 0.93 | 73188 | 65.58 | 1971 | ✓ | ✓ |
| 26.07.2019 | 8.69 | 1.39 | 2.35 | 113310 | 93.26 | 2223 | ✓ | |

| | | | | | | | | |
|---|---|---|---|---|---|---|---|---|
| 11.08.2019 | 6.95 | 1.81 | 1.65 | 88064 | 95.63 | 2323 | | ✓ |
| 20.08.2019 | 0.89 | 1.13 | 0.27 | 6137 | 8.06 | 2031 | ✓ | ✓ |
| 24.06.2021 | 8.18 | 2.40 | 1.69 | 105032 | 162.20 | 1750 | | ✓ |
| 06.07.2021 | 8.69 | 2.50 | 1.75 | 76906 | 186.61 | 1605 | | ✓ |
| 16.07.2021 | 2.78 | 2.38 | 0.58 | 80879 | 60.70 | 1916 | ✓ | ✓ |
| 07.08.2021 | 2.32 | 2.49 | 0.47 | 38737 | 41.19 | 1884 | | ✓ |
| 19.09.2021 | 1.25 | 1.13 | 0.38 | 8538 | 10.67 | 1697 | | ✓ |
| 05.06.2022 | 3.39 | 2.08 | 0.75 | 39498 | 55.42 | 1690 | | ✓ |
| 04.07.2022 | 8.18 | 2.49 | 1.66 | 175929 | 169.14 | 1189 | | ✓ |
| 08.09.2022 | 1.91 | 1.93 | 0.44 | 9283 | 20.94 | 1592 | | ✓ |


*4.2 Numerical modelling with RAMMS*
As mentioned above, we iteratively changed the $\mu$– and $\xi$–friction values upon modelling until
we found a best-fit between the modelled and observed flow velocity and flow depth of each
flow (Appendix C and D). Because the latter properties of a debris flow (velocity and depth)
can be characterized by the Froude number (defined by eq. (*2*)), we first describe the
dependency of the modelled flow pattern on the Froude number, which itself is calculated
using the flow depth and velocity data of the field survey (Table 1). Please note that in this
context, eq. (*2*) predicts that changes in the flow velocity have a larger impact on the Froude
number than variations in flow depth. The simulations showed that RAMMS produces
reasonable results (e.g., Figure 2) for Froude numbers up to about 1.75 (Table 1). For larger
values (e.g., flows with large flow velocities), the simulations predict the occurrence of
standing waves at the debris flow front, which, however, have not been observed at the
Illgraben. Therefore, no simulations were possible for the event on the 26[th] of July 2019,
because this flow was characterized by a Froude number of 2.35. We acknowledge that roll-
waves, which could correspond to the standing waves simulated by RAMMS, do occur in a
debris flow, but such waves are mainly observed in the debris flow body and not at the
bouldery front.

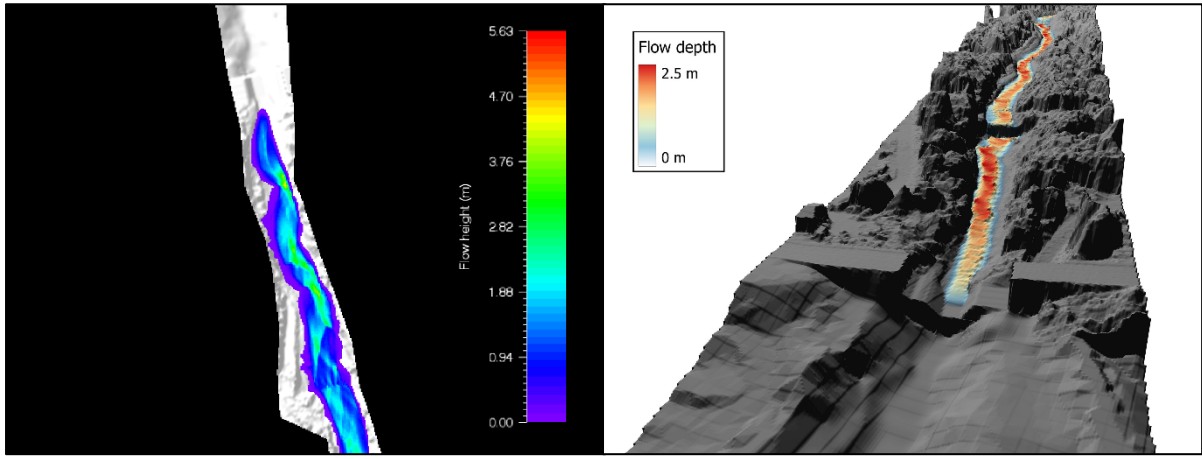

Figure 2: Example of simulated flow depths. The image on the left shows flow depths in a 2D view provided by the
RAMMS software. The image on the right shows a 3D view of the debris flow projected on a hillshade model using
the QGIS software.

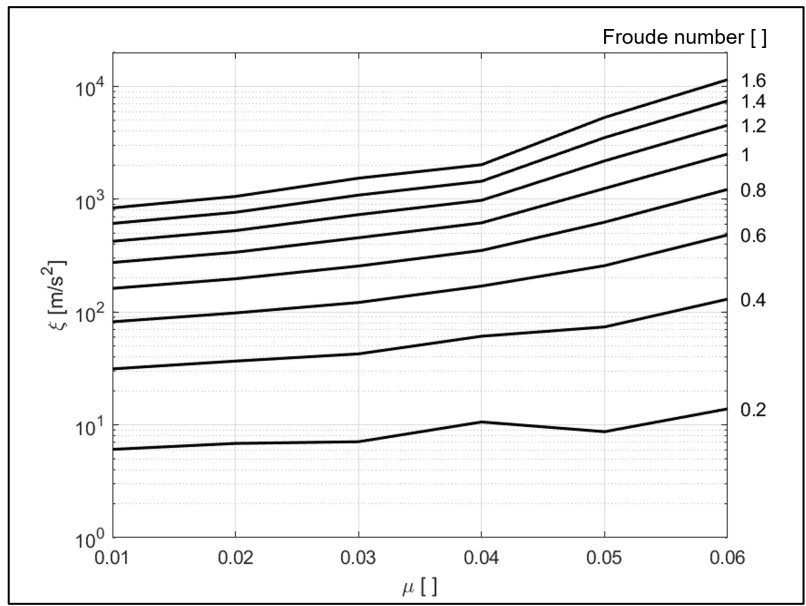

Figure 3: Modelled $\mu$–$\xi$ pairs, which result in a best fit between the observed and the modelled debris flow
parameters. These latter values, in turn, appear to depend on the Froude numbers (based on measurements in
the field, see Table 1) of the corresponding debris flow events.

The model results show that more than one best-fit $\mu$–$\xi$ pair is possible for successfully
reproducing the observed velocity and depth of a flow (Appendix E). Yet, on average, the
lowest $z$-value is calculated for the $\mu$–$\xi$ pair with $\mu$ = 0.01, followed by the pairs with $\mu$ = 0.02
and $\mu$ = 0.05. These values and patterns are consistent with the results of other debris flows
analyses conducted with RAMMS and applied to observations in e.g., the Alps (see Mikoš and
Bezak, 2021, for an overview of related papers), the Himalayas near Luzhuang in China
(Jianjun and Zhang; $\mu$– and $\xi$–values of 0.07 and 1500 m/s$^2$), and the coastal region in the
vicinity of Western Ghats in India (Abraham et al., 2021; $\mu$– and $\xi$–values of 0.01 and 100
m/s$^2$, respectively). In addition, Simoni et al. (2012) found that RAMMS successfully
reproduced the maximum observed runout distances of debris flows in the Italian Alps for $\mu$–
values close to the energy gradient of the debris flow channel (which is the tangent of the
surface slope). Our modelling results support these inferences and additionally show that the
modelled $\mu$– and $\xi$–relationships show a strong dependency on the corresponding Froude
numbers calculated from the field data (Figure 3, and Appendix E, F). Besides, for a given $\mu$–
value, the RAMMS models predict that the $\xi$–values increase with the Froude number. Such
an increase is more obvious for large than for small $\mu$–values (Figure 3, see also Appendix F).

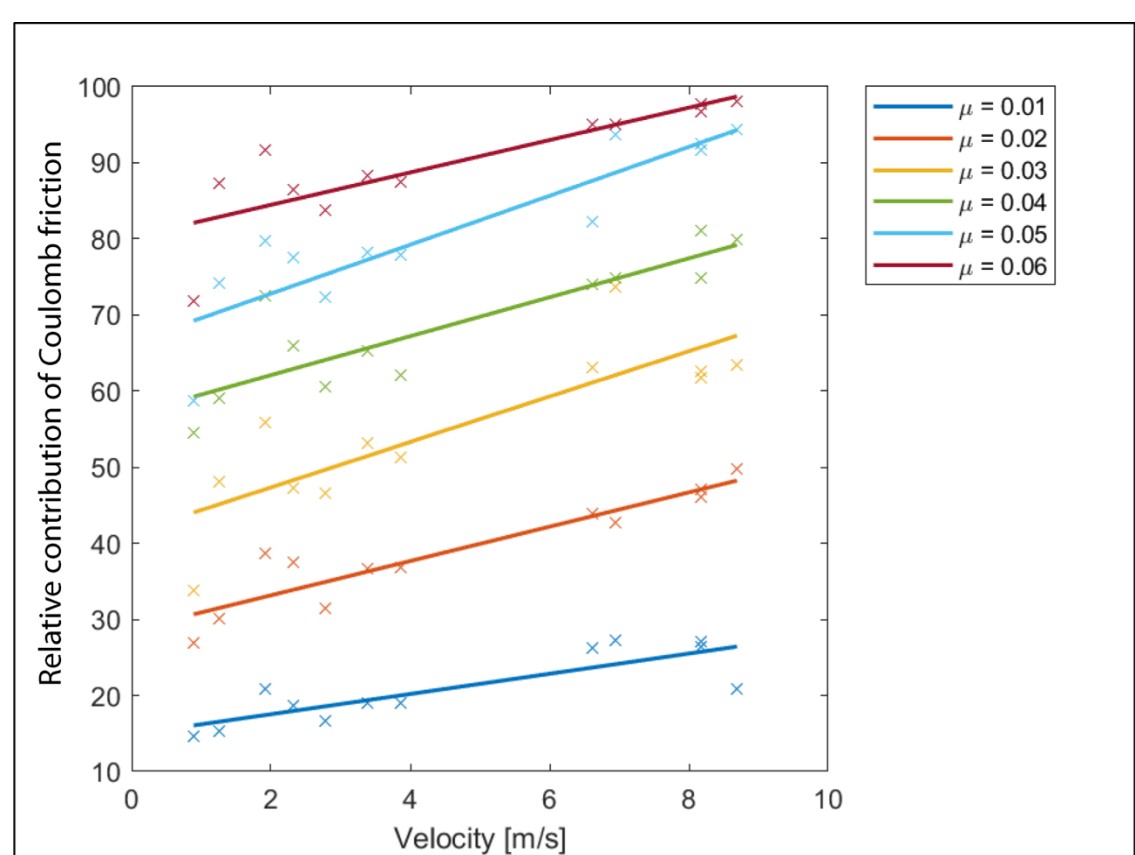

Figure 4: Correlations between the relative contribution of the Coulomb friction to the total friction in
percent (y-axis) and the velocity of a debris flow (x-axis) for different $\mu$-values. The data points refer to
best-fit simulations only. The lines are first order polynomial least square fitted trendlines.
Figure 4 illustrates that upon modelling, the relative contribution of the Coulomb friction to the
total friction increases with the flow velocity, and it shows that this contribution is greater for
large $\mu$-values than for small ones. In particular, while the percentages of the Coulomb friction
are in the range of c. 20% for a $\mu$-value of 0.01 and a flow velocity of < 1 m/s, they increase to
> 90% for a larger $\mu$-value of 0.06 and a flow velocity of > 8 m/s.

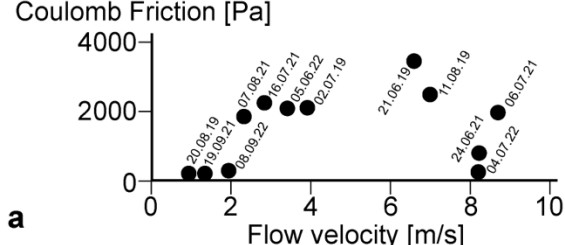
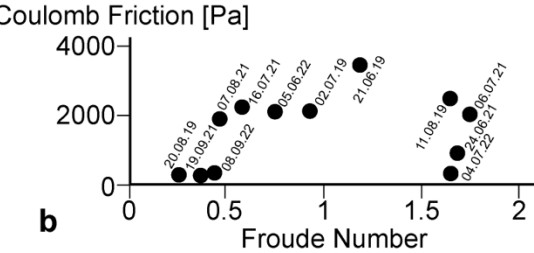

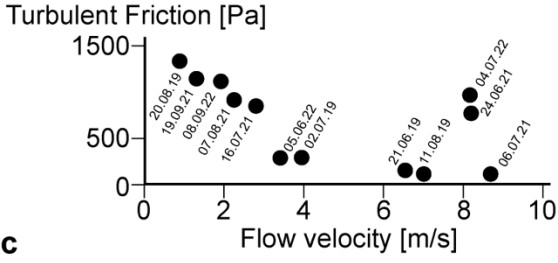
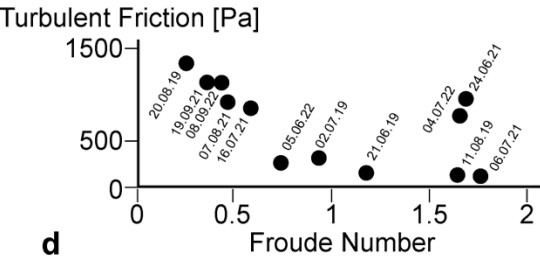

a    b    c    d


Figure 5: Relationships between a) flow velocity and Coulomb friction, b) Froude number and
Coulomb friction, c) flow velocity and turbulent friction, and d) Froude number and turbulent friction.
The plots show the results of the best-fit simulations for each flow.

The relationships elaborated above are further detailed in Figure 5, which documents the dependency of the friction property (Coulomb and turbulent friction) of a flow on its velocity and the corresponding Froude number. Note that for this analysis, we only considered the results of the best-fit simulations of each flow. Accordingly, rapid debris flows with high Froude numbers tend to be characterized by a large Coulomb friction, while slower debris flows are simulated more reliably with a rather low Coulomb friction (Figures 5a, 5b). Conversely, flows with a slow velocity tend to have a larger turbulent friction and are characterized by a low Froude number, whereas the turbulent friction tends to be low for flows with a high velocity and a high Froude number (Figures 5c, 5d). However, we note that the aforementioned relationships between flow velocity, Froude number, turbulent and Coulomb friction break down for flows that are more rapid than c. 6-7 m/s (Figure 5). Similar to the event on 26th of July 2019, these flows are characterized by Froude numbers that are much larger than 1 (Table 1). These flows appear to be in a condition in which the relationships between friction and flow properties are apparently non-linear and more complex than in flows, which can be characterized by low Froude numbers. A further elaboration of this topic is, however, beyond the scope of this paper.

Figure 6 summarizes the consequences of the aforementioned relationships. In particular, small values of the Coulomb friction coefficient ($\mu \sim 0.01$), when the flow is moving on the order of a few meters per second, indicate that the contribution of the Coulomb term to the total friction is small, and that the total friction is therefore dominated by the turbulent friction

term (Figure 6). In the extreme case when $\mu = 0$, the turbulent friction term (eq. *1*) closely
resembles a Chezy friction from open-channel hydraulics (e.g. Henderson, 1966). Large
values of the Coulomb friction coefficient (here $\mu \sim 0.05$) suggest that the Coulomb friction
term is important, and that the contribution of the turbulent friction is correspondingly less
significant (Figure 6). Because we found ideal $\mu$–$\xi$ pairs with $\mu = 0.01$–$0.02$ and $\mu = 0.05$ for
most debris flows, we considered these flows to be dominated either by (i) the turbulent friction
(flows with $\mu = 0.01$–$0.02$) or by (ii) the Coulomb friction (flows with $\mu = 0.05$). Note that Figure
6 also shows that the total friction $S$ increases with a larger $\mu$. Nevertheless, the output of the
simulation (velocity and flow depth) is similar regardless of which $\mu$–$\xi$ pair variant is chosen.

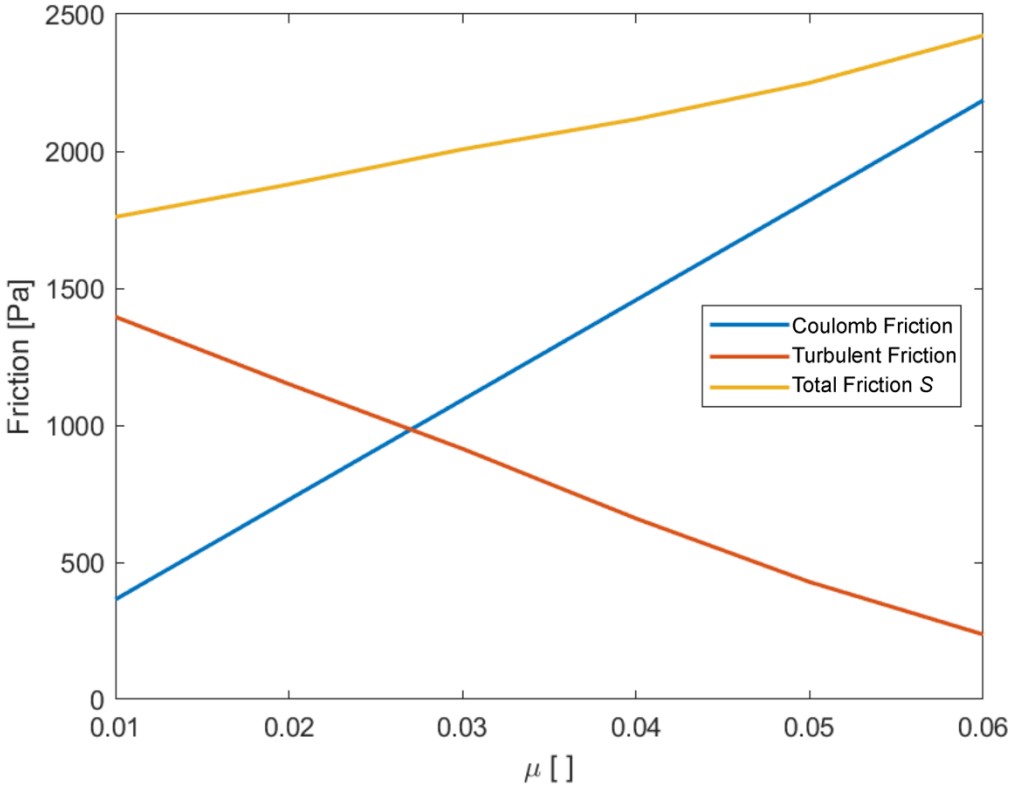


Figure 6: Values of Coulomb friction, turbulent friction, and total friction $S$ as a function of the selected Coulomb
friction coefficient $\mu$. The plotted values are averages of the friction magnitudes of all best-fit simulations. The blue
line represents the Coulomb friction contribution. The red line is the turbulent friction, and the yellow line is the total
friction.

*4.3   Grain size distribution*
Samples from 14 debris flows were analyzed for their grain size distribution (Figure 7 and
Appendix G). Note that there was a sediment sample but no monitoring data for the event on
the 4[th] of October 2021. All events show a very similar grain size distribution. An exception,
and thus an outlier, is a sediment sample that has a larger relative abundance of fine-grained
material. This sample was taken from a debris flow, which occurred on the 2[nd] of July 2019.
For all samples, the clay fraction has a relative mass abundance of 2–3%, the silt fraction 27–
35%, the sand fraction 27–40% and the part of the gravel fraction that is covered by the
analysis 23–37%. Note that the gravel fraction >16 mm was also analyzed (Appendix G). Yet
we normalized the grain size data to 16 mm, because it was not feasible to collect larger mass-
representative samples. Therefore, we acknowledge that the upper percentiles are affected
and thus biased by this cut-off and the related percentage values have to be considered with
caution.
For all samples, we measured grain sizes of 0.015–0.02 mm for the 16% percentile, 4–9 mm
for the 84% percentile and 10–15 mm for the 95% percentile. The median grain size ranges
from 0.15 mm to 0.5 mm. In general, the material was very poorly sorted with a skewness
towards the fine-grained fraction. Interestingly, the grain size distribution was quite similar for
all sampled material. Based on the available datasets, we are neither able to determine
whether the mean grain size is more variable in space than in time, nor can we detect whether
the coarse-grained fraction (>16 mm) could be highly variable whereas the fine-grained
material is more homogeneous. However, similar to the mineralogical composition, which is
also quite similar between the various flows, we interpret that the rather homogeneous
granulometric composition at least of the fine-grained portion of the sediment is the direct
consequence of the cascade of sediment mixing in the upstream part of the Illgraben
(Schlunegger et al., 2009).

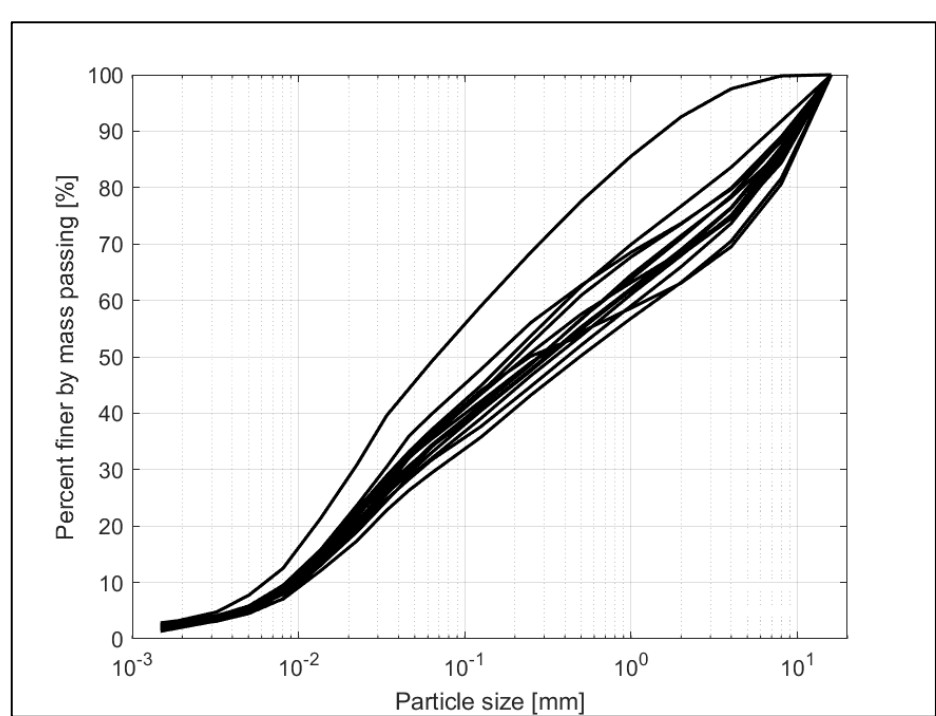


Figure 7: Diagram showing the grain size distribution of all 14 sampled debris flow deposits, truncated at a
maximum grain size of 16 mm.

*4.4    Powder XRD*
The results of the powder XRD analysis (Appendix H) show that quartz was the main mineral
of the silt fraction and contributes between 29 and 36 wt% (Figure 8). In addition, dolomite
(17–24 wt%), muscovite (18–22 wt%), calcite (7–18 wt%) and illite minerals (8–12 wt%) are
present in all samples. Feldspar grains occur by < 5 wt%, and the clay minerals chlorite,
kaolinite and smectite are present in small quantities (< 1%) or are below the detection limit.
Calcite shows the greatest variation in the mineralogical composition with differences up to 11
wt%. The other main components including quartz, dolomite, muscovite and illite show
variations with a maximum of 7 wt%. The feldspar minerals albite and orthoclase are very
homogeneously distributed in the four samples. Overall, the variations in the mineralogical
composition between the different samples are only minor and often lie within the
methodological error of ± 10% of the measured values. Yet, some albeit minor differences can
be detected when the compositions of the coarse- and fine-grained samples are compared.
In the coarse-grained sample, calcite crystals are more abundant than in the sample
characterizing a fine-grained debris flow. In contrast, the latter sample has a larger relative
abundance of illite minerals than the sample made up of coarser sediments. Although the
database is sparse, we tentatively consider these differences to reflect a source signal where
the heavily fractured basement rocks and Triassic schists, which also host the illite crystals,
have the potential to supply larger volumes of fine-grained material than the bedrock made up
of limestones.
From the clay minerals, only smectite can absorb larger amounts of water (Likos and Lu,
2002). However, the x-ray spectra of muscovite and smectite crystals cannot be distinguished
with the applied XRD method. Because the basement rocks and the Triassic schists are
considered to be the source of the clay minerals in the catchment area (Schlunegger et al.,
2009), the signal is more likely related to the fine-grained muscovite (sericite) than to the
smectite minerals (Scheiber et al., 2013). Therefore, swelling clay minerals are expected to
be of minor importance in this case.

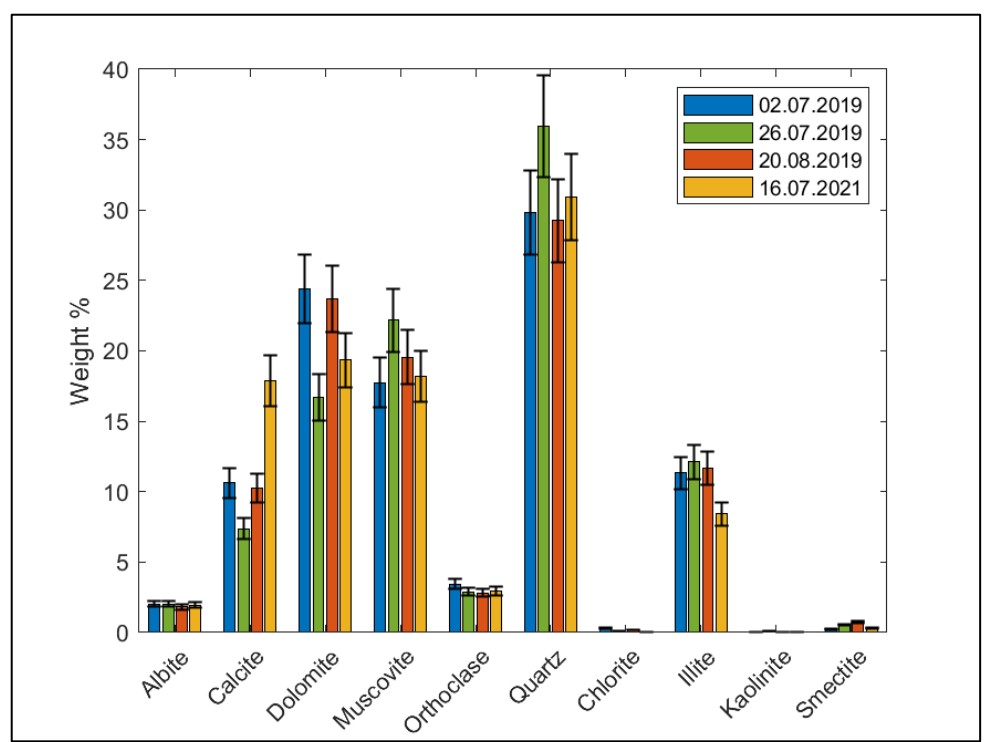

Figure 8: Mineralogic composition of four samples analyzed by powder-XRD. The black error bars indicate a
methodological error of 10% of the measured value. The material representing the flow on the 2nd of July 2019 was
exceptionally fine-grained (Appendix G); the flow on the 26th of July of the same year was an event with a high
velocity (8.69 m/s), and it was the most rapid flow (Table 1). The debris flow on the 20th of August, again in 2019,
was very slow (0.89 m/s) and it was indeed the slowest flow during the survey period (Table 1). The material taken
from the debris flow on the 16th of July 2021 was characterized by a rather coarse-grained matrix (Appendix G).

*4.5    Statistical evaluation of the debris flow properties*
A statistical evaluation of the debris flow parameters measured at the monitoring station shows
a positive correlation between velocity, flow depth, volume, and maximum discharge (Figure
9). While velocity, volume, and maximum discharge correlate very strongly among themselves
as they are physically related (auto-correlation), the correlation of these parameters with the
flow depth is less evident, yet a weak positive correlation is certainly visible. Accordingly, and
as expected (McArdell et al., 2003), a debris flow with a large volume tends to have a large
flow velocity and flow depth, which consequently also results in a large maximum discharge
and a large Froude number. On the other hand, debris flows that have a small volume are
also slow, and they have both a small flow depth and a low Froude number. Interestingly, clear
correlations between grain size, clay content and flow properties are not visible in our analyses
(Figure 10). Also, no correlation between the inferred water content and the volume or
maximum discharge was found for these events. Yet, the total friction values that are extracted
from the modelling results tend to show a positive correlation with the flow depth, and a weak
positive correlation with the density and thus the water content (Figure 9).

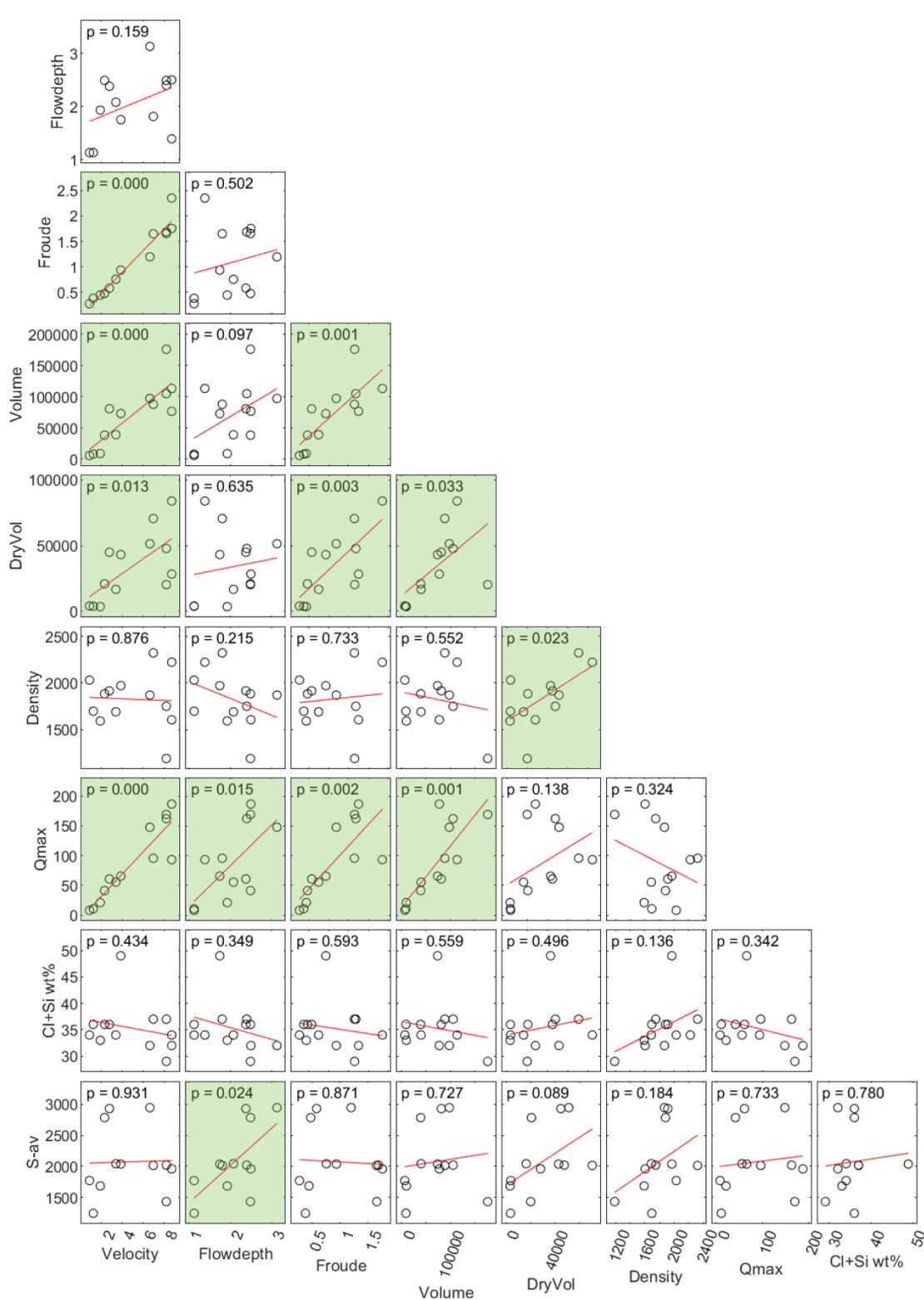

Figure 9: Statistical correlations between dynamic properties of the debris flows with a statistical p-value. For
correlation tests a significance level of 0.05 is considered. Correlations with p-values < 0.05 can therefore be
considered as significant (illustrated with green color). Measurements from the monitoring station at the Illgraben
and values derived from them are front velocity [m/s], maximum flow depth [m], Froude number [ ], total volume
[m$^3$], dry sediment volume [m$^3$], density of a flow [kg/m$^3$], which points to the water content and the maximum
discharge [m³/s]. From the grain size analyses, we have the percentage of the sum of clay and silt in the sample
[wt%]. From the modelling with RAMMS we get the total amount of friction [Pa] as average of all best-fit simulations
of a certain event. The plots were generated using a modified version of the Correlation Matrix Scatterplot by Chow
(2022) for MATLAB. Note that a statistical p-value with p = 0.000 means that the value is less than 0.0005, and
therefore it is rounded down to 0.

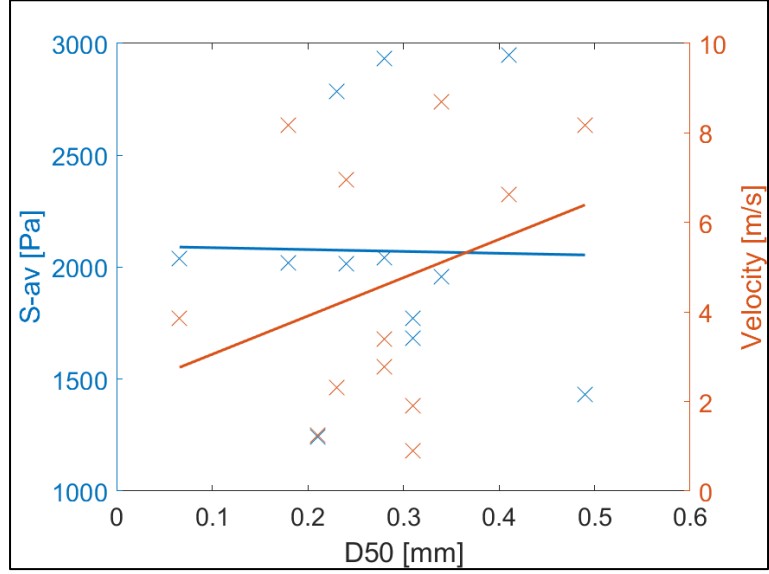

Figure 10: Correlation (first order polynomial trendlines accomplished by least square fitting) between the total
amount of friction S-av as average of all best-fit simulations of a certain event and the D50 value of the
corresponding sediment sample (blue), and correlation between the measured velocity of the flow and the D50
value of the corresponding sediment sample (red).

## 5   Discussion

The debris flows observed in the years 2019, 2021 and 2022 show large differences in their
dynamics, where flow depths and flow velocities varied by a factor of 3 and 10, respectively.
Despite these variabilities in the surveyed parameters, most of the flows could be simulated
with RAMMS, and the model outputs yielded consistent results regarding the underlying
controls and the simulated flow kinematics and properties (see Appendix C, D, E and F, and
the related z-values). In the following section, we discuss how the various parameters such
as the grain size and mineralogical distribution of the fined-grained matrix as well as the friction
properties potentially exerted a control on the surveyed debris flows.

*5.1   Relationships between volume, flow velocity and flow depth, and controls on friction*
*properties*
The statistical tests show positive correlations between volume, flow velocity, flow depth and
maximum discharge rate. Our results are thus consistent with similar results reported by
Rickenmann (1999), de Haas et al. (2015) and Hürlimann et al. (2015) and reflect the open-
channel hydraulic principles used to compute these parameters (McArdell et al., 2023).
Indeed, as shown by the aforementioned authors, flows with larger volumes may contain a
larger number of pebbles and boulders, which according to Johnson et al. (2012) are likely to
accumulate on the front of these flows. As a result, the frictional resistance of the frontal part
increases (Iverson, 1997), with the consequence of a damming effect such as that the flow
depths will increase. We indeed see such a mechanism at work in the surveyed flows through
positive correlations between flow depth, flow velocity and flow volume. We thus infer that the
volume can be considered as the most important driving parameter for explaining the debris
flow dynamics in the Illgraben system and therefore can be considered as a key parameter.
This confirms standard practice in hazard analysis, which gives primary importance to event
volume. We note that this argument relies on the debris flows all having the same initial grain
size distribution, which, as discussed above, we can only document for sediment sizes smaller
than 16 mm. Yet, we acknowledge that a visual comparison of the videos (Appendix B) clearly
shows differences in the abundance of relatively coarse sediment (e.g., boulders). A more
detailed analysis on this topic will require additional data and is beyond the scope of this paper.
The evaluation of the RAMMS simulations shows that there are several $\mu$–$\xi$ pairs, which yield
ideal solutions upon simulating the surveyed debris flows. In particular, the same flow can
successfully be reproduced by RAMMS with large and low Voellmy $\mu$–values. However, an
assessment of which of these possibilities is more appropriate can be found if the flow velocity
is used as a criterion. Indeed, our analysis showed that debris flows with a high velocity (up
to 6-7 m/s) tend to be dominated by a large Coulomb friction (large $\mu$–value), whereas flows
with a low velocity have a low Coulomb friction (low $\mu$–value) but a relatively high turbulent
friction (Figure 5). Yet for flows with velocities that are larger than 6-7 m/s, these relatively
simple relationships break down most likely because such flows appear to be in a condition
where the flow pattern is more complex (e.g., roll waves with Froude numbers that are much
larger than 1 to 1.5, Table 1).
We note that while it is tempting to interpret such low– $\mu$ flows as being 'laminar' and large– $\mu$
flows as 'turbulent' (because of the low and high Froude numbers, see also Figure 5),
independent criteria for determining the presence or absence of turbulence in debris flows are
not yet available. A hydraulics-based estimate based on the Reynolds number to characterize
the presence or absence of turbulence (e.g. Henderson, 1966) requires estimates of the
rheology of the entire flow, which are not available. In addition, it is unclear to what extent
rheological measurements of fine sediment slurries can represent the overall viscosity of the
flow given the presence of other processes such as the jamming of particles in the flow
(Kostynic et al., 2022). Yet, such calculations are beyond the scope of this contribution.
*5.2 Influence of the grain size distribution and mineralogy*
The granulometric analysis of the levee deposits indicates a rather homogeneous grain size
distribution for the clay, silt, sand and the fine-grained gravel fraction. The grain size
distribution fits quite well with the granulometric analyses of the debris flow deposits at the
Illgraben published by Hürlimann et al. (2003). Because of a lack of correlations between the
relative proportion of the fine fraction to the parameters that characterize the flow properties
(e.g., friction and flow velocity; Figure 10), the variations in the dynamics of these flows cannot
be simply explained by a simple fixed friction relation such as in the Voellmy relation. This
inference is consistent with the notion by Iverson (2003) who states that the evolution of debris
flow behavior upstream of the front is likely to be complex. In the same sense, because of the
homogeneity of the samples with respect to the grain size distribution of the components
smaller than 16 mm, also the relative abundance of the sand to the fine-grained gravel fraction
cannot be related to variations in the flow dynamics. Nevertheless, an influence of the grain
size composition on the debris flow dynamics, as described by de Haas et al. (2015) and
Hürlimann et al. (2015), cannot be fully excluded (see section 1.1). Because the relative
abundances of the different fractions are similar, their potential influence on the flow properties
should also be similar for each event. Due to this similarity, such relationships (if present)
would not be detectable with the measurements presented herein. Admittedly, we also have
no information to exclude a potential control of the coarse-grained fraction such as coarse
gravel, cobbles, and boulders, on the flow dynamics, as described by de Haas et al. (2015).
Attempts to reconstruct the full grain size distribution are hampered by a lack of information
on the grain size below the surface of the flow (e.g., Uchida et al., 2021). In addition, the
influence of small changes of the topography on the results was not investigated here, but
could improve the correlations of the flow properties to grain size if adequately considered.
Similar to the grain size distribution of the fine-grained matrix, we do not see a relationship
between the mineralogical composition of the matrix and the flow properties. Among the
various minerals that are present in the debris flow deposits (Appendix H), we expect to see
a control of the sheet silicates on the velocity of the flows, mainly because clay minerals and
particularly smectite-type of clays have the potential to absorb water in their crystal structure
(see section 1.1). We therefore expect that a high relative abundance of such minerals will
alter the flow rheology and particularly the flows' turbulent friction, which is expected to impact
the flow velocity. Apparently, this is not the case at the Illgraben. We consider this absence of
relationships to reflect a supply signal, because the relative abundance of swelling minerals
is negligible in the source area where other sheet silicates such as illite and muscovite crystals
predominate (Scheiber et al., 2013). These silicates don't have swelling properties and
apparently do not impact the velocity of the debris flows at the Illgraben. However, the
homogeneity in terms of the mineral composition and also the grain size composition between
the samples confirms the results of previous studies that inferred the occurrence of an efficient
mixing mechanism as the material is transferred from the source area to the Rhone River
(Schlunegger et al., 2009; Berger et al., 2011a).

**Conclusion**

The results obtained in the Illgraben system by comparing various debris flow parameters with data from runout modelling, grain size analyses, XRD analyses can be summarized as follows:

1) The simulation of debris flows with RAMMS yields multiple solutions with different friction coefficients $\mu$ and $\xi$ in the Voellmy equation. The resulting Coulomb and turbulent friction are correlated with the Froude number and runout velocity of the debris flow yet only as long as the flow velocity is < 6-7 m/s.

2) The dynamics of a debris flow in the Illgraben (i.e., flow velocity and flow depth) is strongly dependent on its volume. If information about the sediment volume in the source area is available, the parameters for simulating a potentially worst-case debris flow and its impact can theoretically be assessed with some uncertainties.

3) Due to the relatively large homogeneity of the deposits with respect to the grain size distribution and the mineralogical composition, an efficient mixing process in the Illgraben can be inferred.

4) Based on these data, variations in the dynamics of different debris flows cannot be attributed to the grain size distributions of the clay, silt, sand or fine-grained gravel fractions. Consequently, an assessment of a potential debris flow or a definition of a simulation based on grain size compositions in the source area is not possible in the case presented here.

Such relationships are particularly useful for the assessment of natural hazards, as they provide specific evidence for the estimation of a debris flow and its impact.

**Acknowledgement**

We are grateful for the technical support provided by Franziska Nyffenegger (grain size analysis), Pierre Lanari and Michael Schwenk (statistics) as well as Frank Gfeller and Anulekha Prasad (XRD analysis). We thank the WSL staff for their support with sampling and the support of Marc Christen and Perry Bartelt (RAMMS) is greatly appreciated.

**Data availability**

All data used in this paper are listed in Table 1 and in the supplementary files.

**Autor contributions**

BM and FS designed the study. DB conducted the experiments, collected the data and processed the samples. DB wrote the paper, with contributions by FS and BM. All authors discussed the article.

**Competing interests**

The authors declare that they have no conflict of interest.

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

## Appendix A: Measurements from monitoring station at the Illgraben

| Event start | Front velocity CD 28-29 (m/s) | Max flow depth laser (m) | Max flow depth radar (m) | Mean bulk density laser (kg/m^3) | Peak velocity (quantile 0.99) laser CD 28-29 (m/s) | Peak velocity (quantile 0.99) radar CD 28-29 (m/s) | Peak discharge (quantile 0.99) laser CD 28-29 (m/s) | Volume laser CD 28-29 (m^3) | Flow duration (min) |
|---|---|---|---|---|---|---|---|---|---|
| 21.06.2019 21:44 | 6.62 | 3.13 | 2.69 | 1870 | 6.55 | 6.57 | 147.61 | 97394 | 43 |
| 02.07.2019 01:26 | 3.86 | 1.75 | 1.73 | 1971 | 5.78 | 5.38 | 65.58 | 73188 | 52 |
| 26.07.2019 19:46 | 8.69 | 1.39 | 1.41 | 2223 | 9.74 | 9.98 | 93.26 | 113310 | 65 |
| 11.08.2019 19:07 | 6.95 | 1.81 | 1.89 | 2323 | 6.90 | 6.91 | 95.63 | 88064 | 88 |
| 20.08.2019 19:03 | 0.89 | 1.13 | 1.10 | 2031 | 1.36 | 1.36 | 8.06 | 6137 | 37 |
| 24.06.2021 17:11 | 8.18 | 2.40 | 2.49 | 1750 | 8.16 | 8.10 | 162.20 | 105032 | 38 |
| 06.07.2021 20:43 | 8.69 | 2.50 | 2.58 | 1605 | 8.65 | 8.67 | 186.61 | 76906 | 28 |
| 16.07.2021 05:43 | 2.78 | 2.38 | 2.44 | 1916 | 3.22 | 3.30 | 60.70 | 80879 | 77 |
| 07.08.2021 16:22 | 2.32 | 2.49 | 2.17 | 1884 | 2.89 | 2.74 | 41.19 | 38737 | 46 |
| 19.09.2021 08:57 | 1.25 | 1.13 | 1.22 | 1697 | 1.41 | 1.39 | 10.67 | 8538 | 43 |
| 05.06.2022 12:33 | 3.39 | 2.08 | 2.15 | 1690 | 4.14 | 4.32 | 55.42 | 39498 | 55 |
| 04.07.2022 22:54 | 8.18 | 2.49 | 2.60 | 1189 | 8.46 | 7.36 | 169.14 | 175929 | 39 |
| 08.09.2022 02:06 | 1.91 | 1.93 | 1.77 | 1592 | 1.85 | 1.87 | 20.94 | 9283 | 20 |

**Appendix B: Video recordings of debris flows. The video camera is placed at the Survey Station (see Figure 1c in main text)**

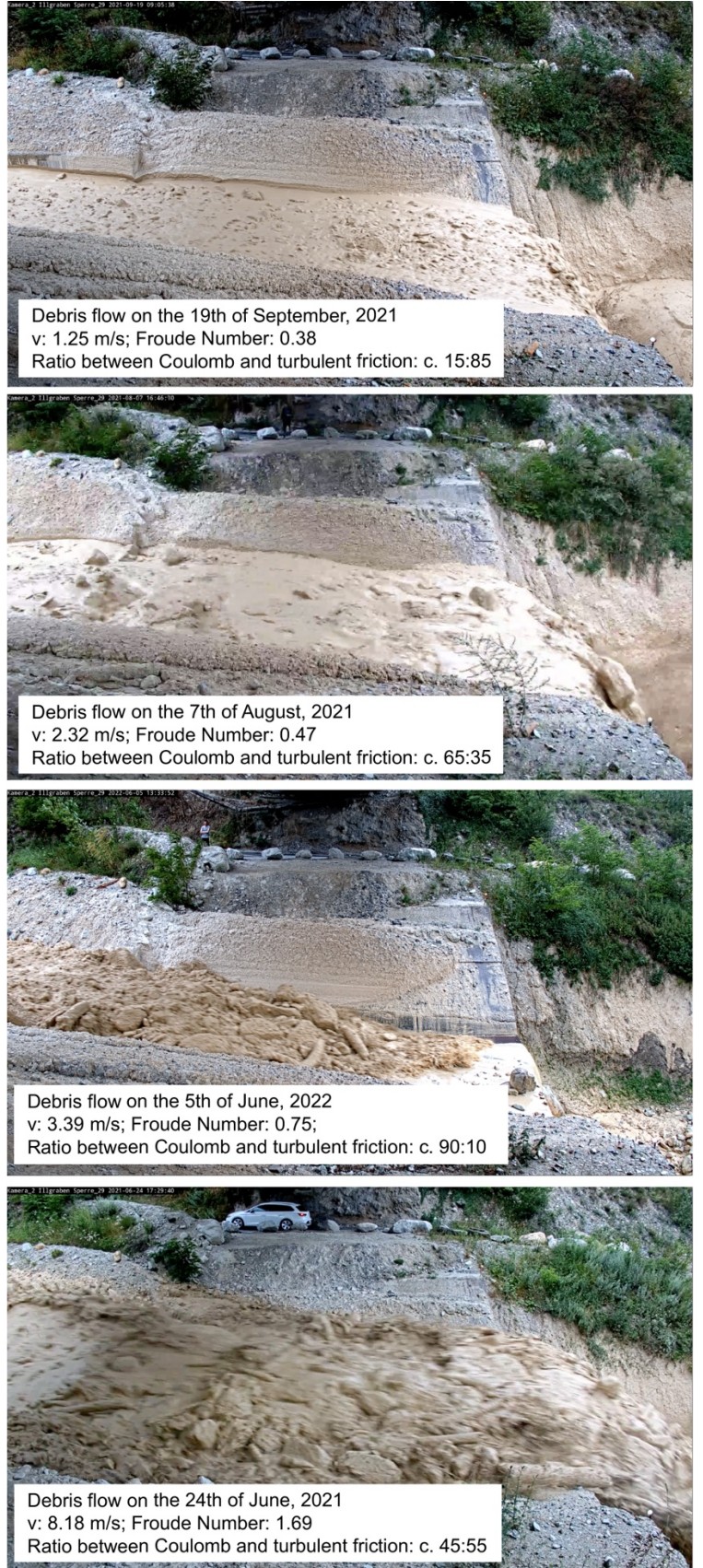

Debris flow on the 19th of September, 2021
v: 1.25 m/s; Froude Number: 0.38
Ratio between Coulomb and turbulent friction: c. 15:85

Debris flow on the 7th of August, 2021
v: 2.32 m/s; Froude Number: 0.47
Ratio between Coulomb and turbulent friction: c. 65:35

Debris flow on the 5th of June, 2022
v: 3.39 m/s; Froude Number: 0.75;
Ratio between Coulomb and turbulent friction: c. 90:10

Debris flow on the 24th of June, 2021
v: 8.18 m/s; Froude Number: 1.69
Ratio between Coulomb and turbulent friction: c. 45:55

## Appendix C: Exemplary evaluation of the simulations of the debris flow event on 24.06.2021

Input data composed of raster and shape files, simulation settings and measurements from the monitoring station.

| Event | 24.06.2021 |
|---|---|
| | |
| DTM | DTM_0.5.tif |
| DTM resolution [m] | 0.5 |
| calculation domain | calcdom.shp |
| release area | hydrograph.shp |
| stop parameter [%] | 5 |
| sim resolution [m] | 0.5 |
| end time [s] | 600 |
| dump step [s] | 2 |
| | |
| erosion layer | erosion.shp |
| erosion density [kg/m3] | 2000 |
| erosion rate [m/s] | 0.025 |
| pot. Erosion depth [per kPa] | 0.1 |
| critical shear stress [kPa] | 1 |
| max erosion depth [m] | 1 |
| | |
| density [kg/m3] | 1750 |
| inflow direction [°] | 60 |
| vol [m3] | 105032 |
| Qmax [m3/s] | 162.2 |
| t1 [s] | 10 |
| v [m/s] | 8.18 |
| | |
| Front velocity CD 28-29 (m/s) | 8.18 |
| Max flow depth laser (m) | 2.4 |
| Max flow depth radar (m) | 2.49 |
| Peak velocity (quantile 0.99) laser CD 28-29 (m/s) | 8.16 |
| Peak velocity (quantile 0.99) radar CD 28-29 (m/s) | 8.1 |
| Flow duration (min) | 38 |
| | |
| CD28-CD29 | 134m |
| CD27-CD29 | 460m |
| | |
| Froude number | 1.69 |

Output data with velocity (v) and flow depth (av_maxd_P). These variables were compared with the results of the field survey to determine the best-fit simulation (green) for each $\mu$. The z-values are calculated from the laser measurement (Max flow depth laser, see above).

| Simulation | Mu [] | Xi [m/s2] | v [m/s] | maxd_P1 [m] | maxd_P2 [m] | maxd_P3 [m] | maxd_P4 [m] | av_maxd_P [m] | Froude number [] | Qmax [m3/s] | z value laser | z value radar |
|---|---|---|---|---|---|---|---|---|---|---|---|---|
| 1 | 0.02 | 1400 | 8.9 | 1.99 | 2.30 | 2.54 | 2.93 | 2.44 | 1.82 | 140 | 0.09 | 0.09 |
| 2 | 0.02 | 800 | 7.9 | 3.00 | 2.94 | 2.84 | 3.03 | 2.95 | 1.47 | 130 | 0.23 | 0.19 |
| 3 | 0.02 | 1000 | 7.9 | 2.63 | 2.71 | 2.62 | 2.89 | 2.71 | 1.53 | 147 | 0.13 | 0.10 |
| 4 | 0.02 | 1200 | 8.4 | 2.22 | 2.38 | 2.40 | 2.71 | 2.43 | 1.72 | 140 | 0.03 | 0.04 |
| 5 | 0.04 | 1500 | 7.9 | 2.94 | 2.86 | 2.65 | 2.86 | 2.83 | 1.50 | 128 | 0.18 | 0.14 |
| 6 | 0.04 | 2000 | 8.4 | 2.60 | 2.64 | 2.57 | 2.69 | 2.63 | 1.66 | 140 | 0.10 | 0.06 |
| 7 | 0.04 | 2500 | 8.4 | 2.45 | 2.55 | 2.54 | 2.78 | 2.58 | 1.67 | 137 | 0.08 | 0.05 |
| 8 | 0.04 | 3000 | 8.4 | 2.04 | 2.55 | 2.69 | 2.97 | 2.56 | 1.68 | 142 | 0.07 | 0.04 |
| 9 | 0.06 | 12000 | 7.9 | 3.21 | 3.12 | 3.00 | 3.12 | 3.11 | 1.43 | 138 | 0.30 | 0.25 |
| 10 | 0.06 | 8000 | 7.9 | 3.31 | 3.22 | 3.20 | 3.50 | 3.31 | 1.39 | 124 | 0.38 | 0.33 |
| 11 | 0.06 | 9000 | 8.4 | 3.38 | 3.26 | 3.16 | 3.39 | 3.30 | 1.48 | 128 | 0.37 | 0.33 |
| 12 | 0.06 | 10000 | 7.9 | 3.19 | 3.09 | 2.97 | 3.22 | 3.12 | 1.43 | 135 | 0.30 | 0.25 |
| 13 | 0.06 | 14000 | 8.4 | 2.62 | 2.59 | 2.67 | 2.95 | 2.71 | 1.63 | 135 | 0.13 | 0.09 |
| 14 | 0.01 | 800 | 8.4 | 2.62 | 2.69 | 2.56 | 2.87 | 2.69 | 1.64 | 138 | 0.12 | 0.08 |
| 15 | 0.01 | 1000 | 8.4 | 2.18 | 2.33 | 2.49 | 2.84 | 2.46 | 1.71 | 138 | 0.04 | 0.03 |
| 16 | 0.01 | 1200 | 8.9 | 1.90 | 2.35 | 2.68 | 3.26 | 2.55 | 1.78 | 142 | 0.11 | 0.09 |
| 17 | 0.03 | 1000 | 7.9 | 2.94 | 2.87 | 2.70 | 3.04 | 2.89 | 1.48 | 139 | 0.21 | 0.16 |
| 18 | 0.03 | 1500 | 8.4 | 2.46 | 2.59 | 2.49 | 2.78 | 2.58 | 1.67 | 144 | 0.08 | 0.05 |
| 19 | 0.03 | 2000 | 8.9 | 2.07 | 2.53 | 2.71 | 2.97 | 2.57 | 1.77 | 145 | 0.11 | 0.09 |
| 20 | 0.03 | 2500 | 9.6 | 1.95 | 2.31 | 2.65 | 3.05 | 2.49 | 1.94 | 136 | 0.18 | 0.17 |
| 21 | 0.04 | 3500 | 8.9 | 1.71 | 2.12 | 2.66 | 3.04 | 2.38 | 1.84 | 136 | 0.09 | 0.10 |
| 22 | 0.05 | 8000 | 8.9 | 1.70 | 2.13 | 2.51 | 2.91 | 2.31 | 1.87 | 137 | 0.10 | 0.11 |
| 23 | 0.05 | 10000 | 8.9 | 2.04 | 2.48 | 2.77 | 3.10 | 2.60 | 1.76 | 137 | 0.12 | 0.10 |
| 24 | 0.05 | 12000 | 8.9 | 1.77 | 2.45 | 2.94 | 3.35 | 2.63 | 1.75 | 136 | 0.13 | 0.10 |
| 25 | 0.05 | 14000 | 8.9 | 1.82 | 2.06 | 2.55 | 3.27 | 2.43 | 1.82 | 136 | 0.09 | 0.09 |
| 26 | 0.05 | 6000 | 8.4 | 2.16 | 2.54 | 2.59 | 2.80 | 2.52 | 1.69 | 146 | 0.06 | 0.03 |
| 27 | 0.06 | 16000 | 8.4 | 2.87 | 2.73 | 2.70 | 2.94 | 2.81 | 1.60 | 133 | 0.17 | 0.13 |
| 28 | 0.04 | 4000 | 8.9 | 1.99 | 2.48 | 2.70 | 3.12 | 2.57 | 1.77 | 139 | 0.11 | 0.09 |
| 29 | 0.02 | 1600 | 7.6 | 2.09 | 2.47 | 2.65 | 3.13 | 2.59 | 1.51 | 139 | 0.10 | 0.08 |
| 30 | 0.02 | 4000 | 11.2 | 1.99 | 1.93 | 2.19 | 2.42 | 2.13 | 2.45 | 152 | 0.39 | 0.40 |
| 31 | 0.04 | 3200 | 8.9 | 1.80 | 2.29 | 2.65 | 3.00 | 2.44 | 1.82 | 135 | 0.09 | 0.09 |
| 32 | 0.05 | 7000 | 8.4 | 1.77 | 2.42 | 2.67 | 2.93 | 2.45 | 1.71 | 139 | 0.03 | 0.03 |
| 33 | 0.06 | 15000 | 7.9 | 2.62 | 2.58 | 2.71 | 3.05 | 2.74 | 1.52 | 137 | 0.15 | 0.11 |

**Appendix D: Details on the modelling approach**

Information on the number of model runs, the intervals between the $\mu-$ and $\xi-$values upon modelling, and event-specific and general input values that were used opon modelling. Appendix B also lists the results of the model runs per event where the model results and observations had a best fit.

**Number of model runs**

| Event | # of simulations | best z-value |
|---|---|---|
| 21.06.19 | 43 | 0.06 |
| 02.07.19 | 34 | 0.32 |
| 11.08.19 | 41 | 0.13 |
| 20.08.19 | 36 | 0.02 |
| 24.06.21 | 37 | 0.03 |
| 06.07.21 | 38 | 0.03 |
| 16.07.21 | 30 | 0.03 |
| 07.08.21 | 23 | 0.23 |
| 19.09.21 | 33 | 0.11 |
| 05.06.22 | 12 | 0.01 |
| 04.07.22 | 13 | 0.02 |
| 08.09.22 | 20 | 0.34 |
| Total | 360 | |

**Variations of $\mu$ and $\xi$ values**

| | | |
|---|---|---|
| $\mu$ | 0.01 | For $\mu$ we only used the values 0.01, 0.02, 0.03, 0.04, 0.05 and 0.06 upon modelling. |
| | | Also upon modeling, the intervals between the $\xi-$values were 1 for those models where we set $\mu$ =1. For larger $\mu-$values, we increased the intervals |
| $\xi$ | 1 to > 1000 | between the subsequent $\xi-$values to >> 1000. We iteratively changed the values until we found a best-fit between model results and observations. |

**Input for RAMMS, which were not event-specific**

| | |
|---|---|
| DTM | DTM_0.5.tif |
| DTM resolution [m] | 0.5 |
| calculation domain | calcdom.shp |
| release area | hydrograph.shp |
| stop parameter [%] | 5 |
| sim resolution [m] | 0.5 |
| end time [s] | 1000 |
| dump step [s] | 2 |
| erosion layer | erosion.shp |
| erosion density [kg/m3] | 2000 |
| erosion rate [m/s] | 0.025 |
| pot. Erosion depth [per kPa] | 0.1 |
| critical shear stress [kPa] | 1 |
| max erosion depth [m] | 1 |
| inflow direction [°] | 60 |
| t1 Hydrograph [s] | 10 |

**Input for RAMMS, which were event-specific**

| Event | 21.06.19 | 02.07.19 | 11.08.19 | 20.08.19 | 24.06.21 | 06.07.21 | 16.07.21 | 07.08.21 | 19.09.21 | 05.06.22 | 04.07.22 | 08.09.22 |
|---|---|---|---|---|---|---|---|---|---|---|---|---|
| density [kg/m$^3$] | 1870 | 1971 | 2323 | 2031 | 1750 | 1605 | 1916 | 1884 | 1697 | 1690 | 1189 | 1592 |
| vol [m$^3$] | 97394 | 73188 | 88064 | 6137 | 105032 | 76906 | 80879 | 38737 | 8538 | 39498 | 175929 | 9283 |
| Qmax [m$^3$/s] | 147.61 | 65.58 | 95.63 | 8.06 | 162.2 | 186.61 | 60.7 | 41.19 | 10.67 | 55.42 | 169.14 | 20.94 |
| Front velocity CD 28-29 [m/s] | 6.62 | 3.86 | 6.95 | 0.89 | 8.18 | 8.69 | 2.78 | 2.32 | 1.25 | 3.39 | 8.18 | 1.91 |
| Max flow depth laser [m] | 3.13 | 1.75 | 1.81 | 1.13 | 2.4 | 2.5 | 2.38 | 2.49 | 1.13 | 2.08 | 2.49 | 1.93 |
| Max flow depth radar [m] | 2.69 | 1.73 | 1.89 | 1.1 | 2.49 | 2.58 | 2.44 | 2.17 | 1.22 | 2.15 | 2.6 | 1.77 |
| Froude Number | 1.19 | 0.93 | 1.65 | 0.27 | 1.69 | 1.75 | 0.58 | 0.47 | 0.38 | 0.75 | 1.66 | 0.44 |

**Best-fit outputs of RAMMs models**

| Event | 21.06.19 | 02.07.19 | 11.08.19 | 20.08.19 | 24.06.21 | 06.07.21 | 16.07.21 | 07.08.21 | 19.09.21 | 05.06.22 | 04.07.22 | 08.09.22 |
|---|---|---|---|---|---|---|---|---|---|---|---|---|
| Front velocity CD 28-29 [m/s] | 6.7 | 3.9 | 7.4 | 0.9 | 8.4 | 8.9 | 2.8 | 2.31 | 1.24 | 3.35 | 8.38 | 1.76 |
| Max flow depth [m] | 2.96 | 2.32 | 2.02 | 1.15 | 2.43 | 2.47 | 2.32 | 1.91 | 1.01 | 2.09 | 2.5 | 1.29 |
| Froude number | 1.24 | 0.83 | 1.66 | 0.27 | 1.72 | 1.81 | 0.59 | 0.53 | 0.39 | 0.74 | 1.69 | 0.49 |
| Qmax [m³/s] | 122 | 54 | 78 | 7 | 140 | 158 | 48 | 35 | 9 | 40 | 143 | 17 |
| $\mu$ | 0.06 | 0.06 | 0.06 | 0.01 | 0.02 | 0.05 | 0.05 | 0.04 | 0.01 | 0.06 | 0.01 | 0.01 |
| $\xi$ | 4500 | 1000 | 8500 | 12 | 1200 | 10000 | 170 | 105 | 25 | 700 | 1000 | 50 |
| z-value | 0.06 | 0.32 | 0.13 | 0.02 | 0.03 | 0.03 | 0.03 | 0.23 | 0.11 | 0.01 | 0.02 | 0.34 |

**Appendix E: Measured and calculated properties for each flow (v, flow depth, Froude number, volume, density), best-fit model results ($\mu$, $\xi$, z) and related total (S) and Coulomb and turbulent frictions**

For each debris flow event, distinct $\mu$-$\xi$ pairs can be used to successfully model the flow properties such as flow velocity and flow depth. The best-fit solutions between model results and observations, characterized by the lowest z-values, are highlighted by the yellow bar. The values of these best-fit results are displayed in Table E.

| Event | v [m/s] | Flow Depth [m] | Froude Number | Volume (m³) | Density (kg/m³) | $\mu$ | $\xi$ | z | Total Friction S [Pa] | Coulomb Friction [Pa] | Turbulent Friction [Pa] | Coulomb Friction [%] | Turbulent Friction [%] |
|---|---|---|---|---|---|---|---|---|---|---|---|---|---|
| 21.06.19 | 6.6 | 3.1 | 1.19 | 97394 | 1870 | 0.01 | 500 | 0.07 | 2166 | 568 | 1598 | 26 | 74 |
| 21.06.19 | 6.6 | 3.1 | 1.19 | 97394 | 1870 | 0.02 | 550 | 0.10 | 2588 | 1135 | 1453 | 44 | 56 |
| 21.06.19 | 6.6 | 3.1 | 1.19 | 97394 | 1870 | 0.03 | 800 | 0.10 | 2702 | 1703 | 999 | 63 | 37 |
| 21.06.19 | 6.6 | 3.1 | 1.19 | 97394 | 1870 | 0.04 | 1000 | 0.11 | 3069 | 2270 | 799 | 74 | 26 |
| 21.06.19 | 6.6 | 3.1 | 1.19 | 97394 | 1870 | 0.05 | 1300 | 0.11 | 3452 | 2838 | 615 | 82 | 18 |
| 21.06.19 | 6.6 | 3.1 | 1.19 | 97394 | 1870 | 0.06 | 4500 | 0.06 | 3583 | 3405 | 178 | 95 | 5 |
| 02.07.19 | 3.9 | 1.8 | 0.93 | 73188 | 1971 | 0.01 | 200 | 0.37 | 1818 | 347 | 1470 | 19 | 81 |
| 02.07.19 | 3.9 | 1.8 | 0.93 | 73188 | 1971 | 0.02 | 250 | 0.37 | 1871 | 695 | 1176 | 37 | 63 |
| 02.07.19 | 3.9 | 1.8 | 0.93 | 73188 | 1971 | 0.03 | 300 | 0.37 | 2022 | 1042 | 980 | 52 | 48 |
| 02.07.19 | 3.9 | 1.8 | 0.93 | 73188 | 1971 | 0.04 | 350 | 0.36 | 2230 | 1389 | 840 | 62 | 38 |
| 02.07.19 | 3.9 | 1.8 | 0.93 | 73188 | 1971 | 0.05 | 600 | 0.35 | 2227 | 1737 | 490 | 78 | 22 |
| **02.07.19** | 3.9 | 1.8 | 0.93 | 73188 | 1971 | 0.06 | 1000 | 0.32 | 2378 | 2084 | 294 | 88 | 12 |
| 11.08.19 | 7 | 1.8 | 1.65 | 88064 | 2323 | 0.01 | 1000 | 0.16 | 1526 | 409 | 1117 | 27 | 73 |
| 11.08.19 | 7 | 1.8 | 1.65 | 88064 | 2323 | 0.02 | 1000 | 0.18 | 1935 | 819 | 1117 | 42 | 58 |
| 11.08.19 | 7 | 1.8 | 1.65 | 88064 | 2323 | 0.03 | 2500 | 0.21 | 1675 | 1228 | 447 | 73 | 27 |
| 11.08.19 | 7 | 1.8 | 1.65 | 88064 | 2323 | 0.04 | 2000 | 0.18 | 2196 | 1637 | 558 | 75 | 25 |
| 11.08.19 | 7 | 1.8 | 1.65 | 88064 | 2323 | 0.05 | 8000 | 0.14 | 2186 | 2047 | 140 | 94 | 6 |
| 11.08.19 | 7 | 1.8 | 1.65 | 88064 | 2323 | 0.06 | 8500 | 0.13 | 2588 | 2456 | 131 | 95 | 5 |
| **20.08.19** | 0.9 | 1.1 | 0.27 | 6137 | 2031 | 0.01 | 12 | 0.02 | 1564 | 219 | 1345 | 14 | 86 |
| 20.08.19 | 0.9 | 1.1 | 0.27 | 6137 | 2031 | 0.02 | 13 | 0.03 | 1679 | 437 | 1241 | 26 | 74 |
| 20.08.19 | 0.9 | 1.1 | 0.27 | 6137 | 2031 | 0.03 | 12 | 0.10 | 2001 | 656 | 1345 | 33 | 67 |
| 20.08.19 | 0.9 | 1.1 | 0.27 | 6137 | 2031 | 0.04 | 21 | 0.12 | 1643 | 875 | 769 | 53 | 47 |
| 20.08.19 | 0.9 | 1.1 | 0.27 | 6137 | 2031 | 0.05 | 20 | 0.17 | 1901 | 1094 | 807 | 58 | 42 |
| 20.08.19 | 0.9 | 1.1 | 0.27 | 6137 | 2031 | 0.06 | 30 | 0.25 | 1850 | 1312 | 538 | 71 | 29 |
| 24.06.21 | 8.2 | 2.4 | 1.69 | 105032 | 1750 | 0.01 | 1000 | 0.04 | 1566 | 411 | 1154 | 26 | 74 |
| 24.06.21 | 8.2 | 2.4 | 1.69 | 105032 | 1750 | 0.02 | 1200 | 0.03 | 1784 | 822 | 962 | 46 | 54 |
| 24.06.21 | 8.2 | 2.4 | 1.69 | 105032 | 1750 | 0.03 | 1500 | 0.08 | 2003 | 1234 | 770 | 62 | 38 |
| 24.06.21 | 8.2 | 2.4 | 1.69 | 105032 | 1750 | 0.04 | 3000 | 0.07 | 2030 | 1645 | 385 | 81 | 19 |
| 24.06.21 | 8.2 | 2.4 | 1.69 | 105032 | 1750 | 0.05 | 7000 | 0.03 | 2221 | 2056 | 165 | 93 | 7 |
| 24.06.21 | 8.2 | 2.4 | 1.69 | 105032 | 1750 | 0.06 | 14000 | 0.13 | 2550 | 2467 | 82 | 97 | 3 |
| 06.07.21 | 8.7 | 2.5 | 1.75 | 76906 | 1605 | 0.01 | 800 | 0.06 | 1883 | 393 | 1490 | 21 | 79 |
| 06.07.21 | 8.7 | 2.5 | 1.75 | 76906 | 1605 | 0.02 | 1500 | 0.06 | 1580 | 786 | 794 | 50 | 50 |
| 06.07.21 | 8.7 | 2.5 | 1.75 | 76906 | 1605 | 0.03 | 1750 | 0.08 | 1860 | 1179 | 681 | 63 | 37 |
| 06.07.21 | 8.7 | 2.5 | 1.75 | 76906 | 1605 | 0.04 | 3000 | 0.09 | 1969 | 1571 | 397 | 80 | 20 |
| **06.07.21** | 8.7 | 2.5 | 1.75 | 76906 | 1605 | 0.05 | 10000 | 0.03 | 2083 | 1964 | 119 | 94 | 6 |
| 06.07.21 | 8.7 | 2.5 | 1.75 | 76906 | 1605 | 0.06 | 25000 | 0.12 | 2405 | 2357 | 48 | 98 | 2 |
| 16.07.21 | 2.8 | 2.4 | 0.58 | 80879 | 1916 | 0.01 | 65 | 0.04 | 2717 | 450 | 2267 | 17 | 83 |
| 16.07.21 | 2.8 | 2.4 | 0.58 | 80879 | 1916 | 0.02 | 75 | 0.04 | 2865 | 900 | 1965 | 31 | 69 |
| 16.07.21 | 2.8 | 2.4 | 0.58 | 80879 | 1916 | 0.03 | 95 | 0.05 | 2902 | 1351 | 1551 | 47 | 53 |
| 16.07.21 | 2.8 | 2.4 | 0.58 | 80879 | 1916 | 0.04 | 125 | 0.07 | 2980 | 1801 | 1179 | 60 | 40 |
| 16.07.21 | 2.8 | 2.4 | 0.58 | 80879 | 1916 | 0.05 | 170 | 0.03 | 3118 | 2251 | 867 | 72 | 28 |
| 16.07.21 | 2.8 | 2.4 | 0.58 | 80879 | 1916 | 0.06 | 280 | 0.04 | 3227 | 2701 | 526 | 84 | 16 |

| | | | | | | | | | | | | | |
|---|---|---|---|---|---|---|---|---|---|---|---|---|---|
| 07.08.21 | 2.3 | 2.5 | 0.47 | 38737 | 1884 | 0.01 | 50 | 0.25 | 2417 | 461 | 1955 | 19 | 81 |
| 07.08.21 | 2.3 | 2.5 | 0.47 | 38737 | 1884 | 0.02 | 65 | 0.26 | 2426 | 922 | 1504 | 38 | 62 |
| 07.08.21 | 2.3 | 2.5 | 0.47 | 38737 | 1884 | 0.03 | 65 | 0.25 | 2888 | 1383 | 1504 | 48 | 52 |
| 07.08.21 | 2.3 | 2.5 | 0.47 | 38737 | 1884 | 0.04 | 105 | 0.23 | 2776 | 1845 | 931 | 66 | 34 |
| 07.08.21 | 2.3 | 2.5 | 0.47 | 38737 | 1884 | 0.05 | 150 | 0.25 | 2957 | 2306 | 652 | 78 | 22 |
| 07.08.21 | 2.3 | 2.5 | 0.47 | 38737 | 1884 | 0.06 | 230 | 0.27 | 3192 | 2767 | 425 | 87 | 13 |
| 19.09.21 | 1.3 | 1.1 | 0.38 | 8538 | 1697 | 0.01 | 25 | 0.11 | 1308 | 183 | 1125 | 14 | 86 |
| 19.09.21 | 1.3 | 1.1 | 0.38 | 8538 | 1697 | 0.02 | 30 | 0.12 | 1303 | 366 | 938 | 28 | 72 |
| 19.09.21 | 1.3 | 1.1 | 0.38 | 8538 | 1697 | 0.03 | 43 | 0.17 | 1203 | 548 | 654 | 46 | 54 |
| 19.09.21 | 1.3 | 1.1 | 0.38 | 8538 | 1697 | 0.04 | 50 | 0.18 | 1294 | 731 | 563 | 57 | 43 |
| 19.09.21 | 1.3 | 1.1 | 0.38 | 8538 | 1697 | 0.05 | 80 | 0.23 | 1265 | 914 | 352 | 72 | 28 |
| 19.09.21 | 1.3 | 1.1 | 0.38 | 8538 | 1697 | 0.06 | 160 | 0.20 | 1272 | 1097 | 176 | 86 | 14 |
| 05.06.22 | 3.4 | 2.1 | 0.75 | 39498 | 1690 | 0.01 | 130 | 0.06 | 1822 | 347 | 1474 | 19 | 81 |
| 05.06.22 | 3.4 | 2.1 | 0.75 | 39498 | 1690 | 0.02 | 160 | 0.05 | 1893 | 695 | 1198 | 37 | 63 |
| 05.06.22 | 3.4 | 2.1 | 0.75 | 39498 | 1690 | 0.03 | 210 | 0.05 | 1955 | 1042 | 913 | 53 | 47 |
| 05.06.22 | 3.4 | 2.1 | 0.75 | 39498 | 1690 | 0.04 | 260 | 0.04 | 2127 | 1390 | 737 | 65 | 35 |
| 05.06.22 | 3.4 | 2.1 | 0.75 | 39498 | 1690 | 0.05 | 400 | 0.03 | 2216 | 1737 | 479 | 78 | 22 |
| 05.06.22 | 3.4 | 2.1 | 0.75 | 39498 | 1690 | 0.06 | 700 | 0.01 | 2359 | 2085 | 274 | 88 | 12 |
| 04.07.22 | 8.2 | 2.5 | 1.66 | 175929 | 1189 | 0.01 | 1000 | 0.02 | 1075 | 291 | 784 | 27 | 73 |
| 04.07.22 | 8.2 | 2.5 | 1.66 | 175929 | 1189 | 0.02 | 1200 | 0.04 | 1236 | 582 | 654 | 47 | 53 |
| 04.07.22 | 8.2 | 2.5 | 1.66 | 175929 | 1189 | 0.03 | 1500 | 0.07 | 1396 | 873 | 523 | 63 | 37 |
| 04.07.22 | 8.2 | 2.5 | 1.66 | 175929 | 1189 | 0.04 | 2000 | 0.09 | 1556 | 1164 | 392 | 75 | 25 |
| 04.07.22 | 8.2 | 2.5 | 1.66 | 175929 | 1189 | 0.05 | 6000 | 0.04 | 1586 | 1455 | 131 | 92 | 8 |
| 04.07.22 | 8.2 | 2.5 | 1.66 | 175929 | 1189 | 0.06 | 20000 | 0.08 | 1785 | 1746 | 39 | 98 | 2 |
| 08.09.22 | 1.9 | 1.9 | 0.44 | 9283 | 1592 | 0.01 | 50 | 0.34 | 1424 | 296 | 1128 | 21 | 79 |
| 08.09.22 | 1.9 | 1.9 | 0.44 | 9283 | 1592 | 0.02 | 60 | 0.36 | 1532 | 592 | 940 | 39 | 61 |
| 08.09.22 | 1.9 | 1.9 | 0.44 | 9283 | 1592 | 0.03 | 80 | 0.37 | 1593 | 888 | 705 | 56 | 44 |
| 08.09.22 | 1.9 | 1.9 | 0.44 | 9283 | 1592 | 0.04 | 125 | 0.39 | 1636 | 1185 | 451 | 72 | 28 |
| 08.09.22 | 1.9 | 1.9 | 0.44 | 9283 | 1592 | 0.05 | 150 | 0.35 | 1857 | 1481 | 376 | 80 | 20 |
| 08.09.22 | 1.9 | 1.9 | 0.44 | 9283 | 1592 | 0.06 | 350 | 0.41 | 1938 | 1777 | 161 | 92 | 8 |

**Appendix F: Best-fit model results per event**

Each debris flow event can be characterized by a distinct $\mu$-$\xi$ pair with a lowest z-value. See Table in Appendix B for best-fit $\mu$-$\xi$ pairs per event.

| Event | v [m/s] | Flow Depth [m] | Froude Number | Volume (m³) | Density (kg/m³) | $\mu$ | $\xi$ | z | Total Friction [Pa] | Coulomb Friction [Pa] | Turbulent Friction [Pa] | Coulomb Friction [%] | Turbulent Friction [%] |
|---|---|---|---|---|---|---|---|---|---|---|---|---|---|
| 21.06.19 | 6.6 | 3.1 | 1.19 | 97394 | 1870 | 0.06 | 4500 | 0.06 | 3583 | 3405 | 178 | 95 | 5 |
| 02.07.19 | 3.9 | 1.8 | 0.93 | 73188 | 1971 | 0.06 | 1000 | 0.32 | 2378 | 2084 | 294 | 88 | 12 |
| 11.08.19 | 7 | 1.8 | 1.65 | 88064 | 2323 | 0.06 | 8500 | 0.13 | 2588 | 2456 | 131 | 95 | 5 |
| 20.08.19 | 0.9 | 1.1 | 0.27 | 6137 | 2031 | 0.01 | 12 | 0.02 | 1564 | 219 | 1345 | 14 | 86 |
| 24.06.21 | 8.2 | 2.4 | 1.69 | 105032 | 1750 | 0.02 | 1200 | 0.03 | 1784 | 822 | 962 | 46 | 54 |
| 06.07.21 | 8.7 | 2.5 | 1.75 | 76906 | 1605 | 0.05 | 10000 | 0.03 | 2083 | 1964 | 119 | 94 | 6 |
| 16.07.21 | 2.8 | 2.4 | 0.58 | 80879 | 1916 | 0.05 | 170 | 0.03 | 3118 | 2251 | 867 | 72 | 28 |
| 07.08.21 | 2.3 | 2.5 | 0.47 | 38737 | 1884 | 0.04 | 105 | 0.23 | 2776 | 1845 | 931 | 66 | 34 |
| 19.09.21 | 1.3 | 1.1 | 0.38 | 8538 | 1697 | 0.01 | 25 | 0.11 | 1308 | 183 | 1125 | 14 | 86 |
| 05.06.22 | 3.4 | 2.1 | 0.75 | 39498 | 1690 | 0.06 | 700 | 0.01 | 2359 | 2085 | 274 | 88 | 12 |
| 04.07.22 | 8.2 | 2.5 | 1.66 | 175929 | 1189 | 0.01 | 1000 | 0.02 | 1075 | 291 | 784 | 27 | 73 |
| 08.09.22 | 1.9 | 1.9 | 0.44 | 9283 | 1592 | 0.01 | 50 | 0.34 | 1424 | 296 | 1128 | 21 | 79 |

# Appendix G: Grain size data

Weight percent passing per mesh size for each sample and complete overview of results from sieving.

| Mesh size [mm] | 21.06.2019 | 02.07.2019 | 26.07.2019 | 11.08.2019 | 20.08.2019 | 24.06.2021 | 06.07.2021 | 16.07.2021 | 07.08.2021 | 19.09.2021 | 04.10.2021 | 05.06.2022 | 04.07.2022 | 08.09.2022 |
|---|---|---|---|---|---|---|---|---|---|---|---|---|---|---|
| 16.0000 | 100.0 | 100.0 | 100.0 | 100.0 | 100.0 | 100.0 | 100.0 | 100.0 | 100.0 | 100.0 | 100.0 | 100.0 | 100.0 | 100.0 |
| 8.0000 | 85.0 | 99.8 | 88.3 | 86.5 | 87.2 | 91.8 | 86.2 | 80.6 | 85.3 | 89.0 | 88.1 | 86.4 | 81.7 | 84.4 |
| 4.0000 | 73.7 | 97.5 | 78.6 | 76.3 | 76.5 | 83.6 | 75.1 | 69.5 | 74.8 | 79.9 | 79.7 | 78.2 | 70.5 | 74.5 |
| 2.0000 | 66.0 | 92.5 | 71.0 | 68.5 | 69.0 | 76.7 | 68.0 | 63.1 | 68.6 | 73.5 | 73.7 | 71.5 | 63.2 | 68.1 |
| 1.0000 | 59.0 | 85.6 | 63.8 | 61.1 | 62.1 | 69.9 | 61.1 | 58.6 | 63.2 | 67.7 | 68.5 | 64.5 | 56.8 | 61.8 |
| 0.5000 | 52.0 | 77.6 | 56.7 | 53.8 | 55.3 | 62.5 | 53.7 | 54.4 | 57.6 | 60.9 | 62.6 | 56.6 | 50.2 | 54.9 |
| 0.2500 | 44.8 | 68.6 | 48.6 | 50.1 | 47.8 | 53.7 | 46.7 | 49.0 | 50.7 | 52.4 | 56.1 | 48.8 | 43.2 | 47.5 |
| 0.1250 | 37.6 | 59.0 | 41.3 | 44.0 | 40.6 | 44.9 | 39.1 | 42.1 | 43.4 | 43.6 | 47.8 | 41.5 | 35.8 | 40.4 |
| 0.0630 | 31.7 | 49.1 | 34.4 | 36.3 | 34.1 | 37.0 | 32.0 | 35.4 | 35.9 | 35.7 | 39.8 | 34.0 | 29.4 | 33.0 |
| 0.0462 | 28.3 | 44.4 | 30.5 | 32.6 | 30.7 | 33.2 | 28.7 | 32.2 | 32.3 | 32.1 | 36.0 | 29.8 | 26.3 | 29.3 |
| 0.0339 | 24.8 | 39.6 | 26.1 | 28.3 | 27.0 | 28.9 | 24.5 | 28.0 | 27.7 | 27.0 | 30.5 | 25.7 | 22.8 | 25.8 |
| 0.0224 | 19.4 | 30.8 | 20.1 | 21.5 | 20.6 | 22.9 | 18.9 | 22.1 | 21.7 | 20.4 | 23.6 | 19.9 | 17.3 | 20.4 |
| 0.0135 | 13.0 | 21.3 | 13.3 | 14.0 | 14.4 | 15.2 | 13.0 | 15.1 | 14.4 | 13.4 | 15.8 | 13.2 | 12.0 | 13.0 |
| 0.0081 | 7.9 | 12.5 | 8.0 | 8.4 | 8.9 | 8.9 | 7.1 | 9.5 | 8.6 | 8.0 | 9.5 | 8.3 | 7.1 | 7.9 |
| 0.0050 | 5.1 | 7.8 | 5.3 | 5.4 | 5.7 | 5.7 | 4.5 | 5.8 | 5.0 | 5.0 | 5.9 | 5.4 | 4.8 | 5.3 |
| 0.0032 | 3.2 | 4.8 | 3.5 | 3.6 | 3.7 | 3.8 | 3.1 | 4.1 | 3.4 | 3.2 | 4.0 | 3.8 | 3.5 | 3.9 |
| 0.0015 | 1.3 | 2.5 | 1.8 | 1.6 | 2.1 | 2.4 | 1.9 | 2.2 | 1.5 | 1.8 | 2.8 | 2.3 | 1.9 | 2.3 |
| 0.0000 | 0.0 | 0.0 | 0.0 | 0.0 | 0.0 | 0.0 | 0.0 | 0.0 | 0.0 | 0.0 | 0.0 | 0.0 | 0.0 | 0.0 |

**Table 1**

| Method | Mesh size [mm] | 21.06.19 Weight [g] | 21.06.19 Weight % passing | 21.06.19 Weight % passing max. 16 mm | 02.07.19 Weight [g] | 02.07.19 Weight % passing | 02.07.19 Weight % passing max. 16 mm | 26.07.19 Weight [g] | 26.07.19 Weight % passing | 26.07.19 Weight % passing max. 16 mm | 11.08.19 Weight [g] | 11.08.19 Weight % passing | 11.08.19 Weight % passing max. 16 mm | 20.08.19 Weight [g] | 20.08.19 Weight % passing | 20.08.19 Weight % passing max. 16 mm |
|---|---|---|---|---|---|---|---|---|---|---|---|---|---|---|---|---|
| Event / Sample mass [g] | | 1958.7 | | | 1772.4 | | | 2856.1 | | | 3299.7 | | | 3001.5 | | |
| dry sieving | 125.0000 | 0.0 | 100.0 | 100.0 | 0.0 | 100.0 | 100.0 | 0.0 | 100.0 | 100.0 | 0.0 | 100.0 | 100.0 | 0.0 | 100.0 | 100.0 |
| | 63.0000 | 0.0 | 100.0 | 100.0 | 0.0 | 100.0 | 100.0 | 0.0 | 100.0 | 100.0 | 0.0 | 100.0 | 100.0 | 0.0 | 100.0 | 100.0 |
| | 31.5000 | 0.0 | 100.0 | 100.0 | 0.0 | 100.0 | 100.0 | 55.7 | 98.1 | 100.0 | 0.0 | 100.0 | 100.0 | 0.0 | 100.0 | 100.0 |
| | 16.0000 | 365.0 | 81.4 | 100.0 | 0.0 | 100.0 | 100.0 | 158.9 | 92.5 | 100.0 | 472.6 | 85.7 | 100.0 | 327.9 | 89.1 | 100.0 |
| | 8.0000 | 239.7 | 69.1 | 85.0 | 3.4 | 99.8 | 99.8 | 308.3 | 81.7 | 88.3 | 380.6 | 74.1 | 86.5 | 341.7 | 77.7 | 87.2 |
| | 4.0000 | 180.2 | 59.9 | 73.7 | 40.5 | 97.5 | 97.5 | 258.1 | 72.7 | 78.6 | 288.3 | 65.4 | 76.3 | 287.4 | 68.1 | 76.5 |
| | 2.0000 | 122.5 | 53.7 | 66.0 | 88.3 | 92.5 | 92.5 | 199.3 | 65.7 | 71.0 | 220.4 | 58.7 | 68.5 | 198.8 | 61.5 | 69.0 |
| | 1.0000 | 110.5 | 48.0 | 59.0 | 123.6 | 85.6 | 85.6 | 189.9 | 59.0 | 63.8 | 211.7 | 52.3 | 61.1 | 184.7 | 55.3 | 62.1 |
| | 0.5000 | 111.4 | 42.3 | 52.0 | 142.0 | 77.6 | 77.6 | 187.3 | 52.5 | 56.7 | 206.6 | 46.1 | 53.8 | 182.5 | 49.3 | 55.3 |
| wet sieving | 0.2500 | | 36.4 | 44.8 | | 68.6 | 68.6 | | 45.0 | 48.6 | | 42.9 | 50.1 | | 42.6 | 47.8 |
| | 0.1250 | | 30.6 | 37.6 | | 59.0 | 59.0 | | 38.2 | 41.3 | | 37.7 | 44.0 | | 36.2 | 40.6 |
| | 0.0630 | | 25.8 | 31.7 | | 49.1 | 49.1 | | 31.8 | 34.4 | | 31.1 | 36.3 | | 30.4 | 34.1 |
| slurry test | 0.0462 | | 23.0 | 28.3 | | 44.4 | 44.4 | | 28.2 | 30.5 | | 27.9 | 32.6 | | 27.4 | 30.7 |
| | 0.0339 | | 20.2 | 24.8 | | 39.6 | 39.6 | | 24.2 | 26.1 | | 24.2 | 28.3 | | 24.0 | 27.0 |
| | 0.0224 | | 15.8 | 19.4 | | 30.8 | 30.8 | | 18.6 | 20.1 | | 18.4 | 21.5 | | 18.3 | 20.6 |
| | 0.0135 | | 10.6 | 13.0 | | 21.3 | 21.3 | | 12.3 | 13.3 | | 12.0 | 14.0 | | 12.8 | 14.4 |
| | 0.0081 | | 6.4 | 7.9 | | 12.5 | 12.5 | | 7.4 | 8.0 | | 7.2 | 8.4 | | 7.9 | 8.9 |
| | 0.0050 | | 4.1 | 5.1 | | 7.8 | 7.8 | | 4.9 | 5.3 | | 4.6 | 5.4 | | 5.1 | 5.7 |
| | 0.0032 | | 2.6 | 3.2 | | 4.8 | 4.8 | | 3.2 | 3.5 | | 3.1 | 3.6 | | 3.3 | 3.7 |
| | 0.0015 | | 1.1 | 1.3 | | 2.5 | 2.5 | | 1.7 | 1.8 | | 1.4 | 1.6 | | 1.9 | 2.1 |
| | 0.0000 | | 0.0 | 0.0 | | 0.0 | 0.0 | | 0.0 | 0.0 | | 0.0 | 0.0 | | 0.0 | 0.0 |

**Table 2**

| Method | Mesh size [mm] | 24.06.21 Weight [g] | 24.06.21 Weight % passing | 24.06.21 Weight % passing max. 16 mm | 06.07.21 Weight [g] | 06.07.21 Weight % passing | 06.07.21 Weight % passing max. 16 mm | 16.07.21 Weight [g] | 16.07.21 Weight % passing | 16.07.21 Weight % passing max. 16 mm | 07.08.21 Weight [g] | 07.08.21 Weight % passing | 07.08.21 Weight % passing max. 16 mm | 19.09.21 Weight [g] | 19.09.21 Weight % passing | 19.09.21 Weight % passing max. 16 mm |
|---|---|---|---|---|---|---|---|---|---|---|---|---|---|---|---|---|
| Event / Sample mass [g] | | 2652.5 | | | 3341.9 | | | 2511.2 | | | 2965.8 | | | 2553.6 | | |
| dry sieving | 125.0000 | 0.0 | 100.0 | 100.0 | 0.0 | 100.0 | 100.0 | 0.0 | 100.0 | 100.0 | 0.0 | 100.0 | 100.0 | 0.0 | 100.0 | 100.0 |
| | 63.0000 | 0.0 | 100.0 | 100.0 | 0.0 | 100.0 | 100.0 | 0.0 | 100.0 | 100.0 | 0.0 | 100.0 | 100.0 | 0.0 | 100.0 | 100.0 |
| | 31.5000 | 0.0 | 100.0 | 100.0 | 296.0 | 83.3 | 100.0 | 434.0 | 84.8 | 100.0 | 602.3 | 81.7 | 100.0 | 212.2 | 92.9 | 100.0 |
| | 16.0000 | 102.7 | 94.8 | 100.0 | 290.2 | 66.9 | 100.0 | 615.7 | 63.2 | 100.0 | 795.1 | 57.7 | 100.0 | 405.6 | 79.4 | 100.0 |
| | 8.0000 | 152.8 | 87.0 | 91.8 | 163.4 | 57.7 | 86.2 | 349.9 | 51.0 | 80.6 | 279.6 | 49.2 | 85.3 | 261.1 | 70.7 | 89.0 |
| | 4.0000 | 151.6 | 79.2 | 83.6 | 131.8 | 50.3 | 75.1 | 200.7 | 44.0 | 69.5 | 199.4 | 43.1 | 74.8 | 217.9 | 63.5 | 79.9 |
| | 2.0000 | 127.9 | 72.7 | 76.7 | 84.7 | 45.5 | 68.0 | 116.3 | 39.9 | 63.1 | 119.0 | 39.5 | 68.6 | 151.8 | 58.4 | 73.5 |
| | 1.0000 | 125.9 | 66.3 | 69.9 | 81.8 | 40.9 | 61.1 | 80.2 | 37.1 | 58.6 | 102.7 | 36.4 | 63.2 | 139.3 | 53.8 | 67.7 |
| | 0.5000 | 138.8 | 59.2 | 62.5 | 87.4 | 35.9 | 53.7 | 76.7 | 34.4 | 54.4 | 106.6 | 33.2 | 57.6 | 162.5 | 48.3 | 60.9 |
| wet sieving | 0.2500 | | 50.9 | 53.7 | | 31.2 | 46.7 | | 31.0 | 49.0 | | 29.3 | 50.7 | | 41.6 | 52.4 |
| | 0.1250 | | 42.5 | 44.9 | | 26.2 | 39.1 | | 26.6 | 42.1 | | 25.0 | 43.4 | | 34.6 | 43.6 |
| | 0.0630 | | 35.1 | 37.0 | | 21.4 | 32.0 | | 22.4 | 35.4 | | 20.7 | 35.9 | | 28.3 | 35.7 |
| slurry test | 0.0462 | | 31.5 | 33.2 | | 19.2 | 28.7 | | 20.3 | 32.2 | | 18.6 | 32.3 | | 25.5 | 32.1 |
| | 0.0339 | | 27.4 | 28.9 | | 16.4 | 24.5 | | 17.7 | 28.0 | | 15.9 | 27.7 | | 21.5 | 27.0 |
| | 0.0224 | | 21.7 | 22.9 | | 12.7 | 18.9 | | 14.0 | 22.1 | | 12.5 | 21.7 | | 16.2 | 20.4 |
| | 0.0135 | | 14.4 | 15.2 | | 8.7 | 13.0 | | 9.5 | 15.1 | | 8.3 | 14.4 | | 10.7 | 13.4 |
| | 0.0081 | | 8.4 | 8.9 | | 4.8 | 7.1 | | 6.0 | 9.5 | | 5.0 | 8.6 | | 6.4 | 8.0 |
| | 0.0050 | | 5.4 | 5.7 | | 3.0 | 4.5 | | 3.6 | 5.8 | | 2.9 | 5.0 | | 4.0 | 5.0 |
| | 0.0032 | | 3.6 | 3.8 | | 2.1 | 3.1 | | 2.6 | 4.1 | | 1.9 | 3.4 | | 2.5 | 3.2 |
| | 0.0015 | | 2.3 | 2.4 | | 1.3 | 1.9 | | 1.4 | 2.2 | | 0.9 | 1.5 | | 1.4 | 1.8 |
| | 0.0000 | | 0.0 | 0.0 | | 0.0 | 0.0 | | 0.0 | 0.0 | | 0.0 | 0.0 | | 0.0 | 0.0 |

**Table 3**

| Method | Mesh size [mm] | 04.10.21 Weight [g] | 04.10.21 Weight % passing | 04.10.21 Weight % passing max. 16 mm | 05.06.22 Weight [g] | 05.06.22 Weight % passing | 05.06.22 Weight % passing max. 16 mm | 04.07.22 Weight [g] | 04.07.22 Weight % passing | 04.07.22 Weight % passing max. 16 mm | 08.09.22 Weight [g] | 08.09.22 Weight % passing | 08.09.22 Weight % passing max. 16 mm |
|---|---|---|---|---|---|---|---|---|---|---|---|---|---|
| Event / Sample mass [g] | | 2788.3 | | | 2866.9 | | | 2677.2 | | | 3400.6 | | |
| dry sieving | 125.0000 | 0.0 | 100.0 | 100.0 | 0.0 | 100.0 | 100.0 | 0.0 | 100.0 | 100.0 | 0.0 | 100.0 | 100.0 |
| | 63.0000 | 0.0 | 100.0 | 100.0 | 0.0 | 100.0 | 100.0 | 0.0 | 100.0 | 100.0 | 0.0 | 100.0 | 100.0 |
| | 31.5000 | 147.8 | 92.5 | 100.0 | 116.2 | 93.4 | 100.0 | 315.5 | 89.0 | 100.0 | 167.8 | 94.9 | 100.0 |
| | 16.0000 | 284.9 | 77.9 | 100.0 | 312.6 | 75.8 | 100.0 | 553.6 | 69.6 | 100.0 | 527.1 | 78.9 | 100.0 |
| | 8.0000 | 180.9 | 68.7 | 88.1 | 182.8 | 65.5 | 86.4 | 364.6 | 56.8 | 81.7 | 406.8 | 66.6 | 84.4 |
| | 4.0000 | 128.7 | 62.1 | 79.7 | 109.5 | 59.3 | 78.2 | 221.3 | 49.1 | 70.5 | 256.4 | 58.8 | 74.5 |
| | 2.0000 | 92.4 | 57.4 | 73.7 | 90.7 | 54.2 | 71.5 | 145.4 | 44.0 | 63.2 | 168.7 | 53.7 | 68.1 |
| | 1.0000 | 78.2 | 53.4 | 68.5 | 93.5 | 48.9 | 64.5 | 126.4 | 39.5 | 56.8 | 162.7 | 48.8 | 61.8 |
| | 0.5000 | 89.8 | 48.8 | 62.6 | 106.9 | 42.9 | 56.6 | 132.2 | 34.9 | 50.2 | 181.1 | 43.3 | 54.9 |
| wet sieving | 0.2500 | | 43.7 | 56.1 | | 37.0 | 48.8 | | 30.1 | 43.2 | | 37.5 | 47.5 |
| | 0.1250 | | 37.2 | 47.8 | | 31.5 | 41.5 | | 24.9 | 35.8 | | 31.9 | 40.4 |
| | 0.0630 | | 31.0 | 39.8 | | 25.8 | 34.0 | | 20.4 | 29.4 | | 26.1 | 33.0 |
| slurry test | 0.0462 | | 28.0 | 36.0 | | 22.6 | 29.8 | | 18.3 | 26.3 | | 23.1 | 29.3 |
| | 0.0339 | | 23.8 | 30.5 | | 19.5 | 25.7 | | 15.9 | 22.8 | | 20.4 | 25.8 |
| | 0.0224 | | 18.4 | 23.6 | | 15.1 | 19.9 | | 12.1 | 17.3 | | 16.1 | 20.4 |
| | 0.0135 | | 12.3 | 15.8 | | 10.0 | 13.2 | | 8.4 | 12.0 | | 10.3 | 13.0 |
| | 0.0081 | | 7.4 | 9.5 | | 6.3 | 8.3 | | 4.9 | 7.1 | | 6.2 | 7.9 |
| | 0.0050 | | 4.6 | 5.9 | | 4.1 | 5.4 | | 3.4 | 4.8 | | 4.2 | 5.3 |
| | 0.0032 | | 3.1 | 4.0 | | 2.9 | 3.8 | | 2.5 | 3.5 | | 3.0 | 3.9 |
| | 0.0015 | | 2.2 | 2.8 | | 1.8 | 2.3 | | 1.3 | 1.9 | | 1.8 | 2.3 |
| | 0.0000 | | 0.0 | 0.0 | | 0.0 | 0.0 | | 0.0 | 0.0 | | 0.0 | 0.0 |

**Appendix H: Results of powder x-ray diffraction analysis**

Measured weight percent per mineral for all four analyzed samples.

| | 02.07.2019 | 26.07.2019 | 20.08.2019 | 16.07.2021 |
|---|---|---|---|---|
| Albite | 2.0 | 2.0 | 1.8 | 2.0 |
| Calcite | 10.6 | 7.4 | 10.2 | 17.9 |
| Dolomite | 24.4 | 16.7 | 23.7 | 19.4 |
| Muscovite | 17.7 | 22.2 | 19.6 | 18.2 |
| Orthoclase | 3.4 | 2.9 | 2.8 | 3.0 |
| Quartz | 29.9 | 36.0 | 29.2 | 30.9 |
| Chlorite | 0.3 | 0.1 | 0.2 | 0.0 |
| Illite | 11.3 | 12.1 | 11.7 | 8.4 |
| Kaolinite | 0.0 | 0.1 | 0.1 | 0.0 |
| Smektite | 0.2 | 0.6 | 0.7 | 0.3 |