# Peer review of "Comparison of debris-flow observations, including fine sediment grain size and composition, and runout model results at the Illgraben, Swiss Alps"

_Natural Hazards and Earth System Sciences, 2023_

## Author Response (AR1)

*Our comment: We greatly appreciate the detailed and careful review by referee 1. Many thanks!*

General comment

This study investigated relationships between flow parameters and the debris flow dynamics using the monitoring data in Illgraben. Friction parameters in RAMMS, which had strong dependency on the Froude number, were used to discuss flow characteristics. Sediment samples from levee deposits were analyzed for their grain size distribution and mineralogical properties. The relationship between Froude number and friction parameters is interesting. The correlations between dynamic properties of debris flows in Figure 8 are important to understand characteristics of real debris flows. Structure of the paper is organized well. The manuscript is acceptable after minor revisions. My major comments are as follows;
      • The total frictional resistance S was compared with dynamic properties of debris flows in Fig. 9. Is it possible to compare basal and internal frictions with the dynamic properties? The comparison would support the discussion in lines 432 to 439.

*Our response: First, we will change the term basal friction to Coulomb friction, and internal friction to turbulent friction as these are the proper terms for the Voellmy equation. We will then show new Figures, which show the relative contribution of the Coulomb friction to the total friction for different friction coefficients µ (new Figure 4), and we will document how this influences the flow velocities and thus the flow properties (new Figure 5). The strong dependence of the relative amount of basal friction on the choice of the coefficient µ as described above can be verified by looking at the different trendlines represented by the different µ-values. While the percentages of Coulomb friction are in the range of c. 20% for a µ-value of 0.01, they increase to around 90% for a higher µ-value of 0.06. In an additional new Figure 5, we will also show a positive correlation between the relative amount of the Coulomb friction, the velocity of the debris flow for each best-fit simulation, and the related Froude numbers. This means that best-fit simulations for fast debris flows tend to be obtained with higher percentages of Coulomb friction, while slower debris flows are simulated more reliably with a rather low Coulomb friction value. Accordingly, flows with a low velocity tend to have a higher turbulent friction and are characterized by a low Froude number, where the turbulent friction tends to be low for flows with a high velocity and a high Froude number (new Figures 5b, 5c).*
*However, we also find (new Figure 5) that these relationships between flow velocity, Froude number, turbulent and Coulomb friction break down for flows that propagate more rapidly than c. 6-7 m/s. Similar to the event on 26th of July 2019, these flows are characterized by Froude numbers that are much larger than 1. These flows might be in a flow state where linear relationships between friction properties and flow properties are likely to break down.*

      • Debris flows were classified into laminar and turbulent flows in line 312 to 314. Is there any relationship between flow type (laminar and turbulent flows) and frictional resistance (basal, internal, and total frictions)? This also supports the discussion in lines 432-439.
*Our response: We will indeed show a new Figure 5 where the relationships between the friction properties, the flow velocity and the Froude number will be displayed, and we will modify the text accordingly. See also response above.*

      • I understand that the monitoring methods in Illgraben have been explained in published papers. However, brief explanations of the monitoring methods, such as flow velocity, flow stage, and discharge are needed, because these are basic data in this study.
*Our response: This will be done. The most relevant information on the methodology are presented in Hürlimann et al. (2003), McArdell et al. (2007) and Schlunegger et al. (2009), and all survey results are listed in the openly accessible database by McArdell et al. (2023). We present a brief summary of the major points in the revised article.*

Specific comments
Title: "Modeling" and "grain size" looks like key words in the title. However, this study is not developing the model by considering grain size. Additionally, this study could not find clear relationship between the grain size and runout characteristics. I suggest authors to change the title of this manuscript.
*Our response: This will be done. We will change the title to better outline what we have done.*

Line 27: Is "(ii)" needed here?
*Our response: No, we will remove it.*

Line 74: There are studies that suspension of fine particles increases debris flow mobility (Iverson, 1997; Kaitna et al., 2016; Uchida et al., 2020). Please add explains on positive aspect of fine particles on debris flow mobility.
Kaitna, R., Palucis, M. C., Yohannes, B., Hill, K. M., and Dietrich, W. E.: Effects of coarse grain size distribution and fine particle content on pore fluid pressure and shear behavior in experimental debris flows, J. Geophys. Res. Earth Surf., 121, 415–441, doi:10.1002/2015JF003725, 2016.

*Our response: Thanks for mentioning this. We gladly refer to these papers in the related context.*

Line 143: two distinct sediment sources Please describe specific sources clearly.
*Our response: This has been done.*

Line 221
Could you show μ, ξ, and z values for each events? Difference in μ and ξ values among debris flow events could be interest of readers. Z value is important to evaluate how RAMMS appropriately simulated debris flows.
*Our response. Yes indeed. These data will be shown in a new Appendix E and F.*

Line 224: 4.3 Grain size distribution
Although maximum grain size in the analysis was 16 mm, it is better to clarify that there are sediments larger than 16 mm in levee deposits.
*Our response: Yes, we mention this at the end of the corresponding chapter.*

Lin 244: four samples were chosen…
In Fig. 7, which samples were from fast and slow velocity flows? Which were coarse-grained and fine-grained samples?
*Our response: We mention this in the caption of this Figure. Thanks for noting.*

Line 245: Authors sampled deposits where coarse-grained or the fine-grained fractions dominate. However, grain size distribution showed that there was no significant change in the grain size amongst events (Fig. 6). Do you mean grain size is more variable depending on space rather than time? Or do you that mean coarse particles are variable but fine particles are homogeneous?
*Our response: Yes indeed, the grain size distribution was quite similar for all sampled material. Based on the available datasets, we are currently not able to identify whether the mean grain size is more variable in space than in time, or if the coarse-grained fraction (>16 mm) could be highly variable, whereas the fine-grained material is more homogeneous, as noted by the reviewer. However, similar to the mineralogical composition, which is also quite similar between the various flows, we interpret that the rather homogeneous granulometric composition at least of the fine-grained portion of the sediment is the direct consequence of the cascade of sediment mixing in the upstream part of the Illgraben (Schlunegger et al., 2009).*

Line 269: the "176.000 m3" should be "176,000 m3"
*Our response: Corrected*

Lines 283-285: Please refer Table 1 in this sentence.
*Our response: Done*

Line 298: Again, it is helpful to understand this sentence if you show μ, ξ, and z values in a table.
*Our response: We add a new Appendix E, which will show this information. See also the response above.*

Lines 385-387: I could not understand relationship between this sentence and Fig. 9.
*Our response: This has been corrected. The text refers to the wrong figure. Thanks for noting.*

Lines 408-410: It is not clear if the debris flows could be nicely simulated by RAMMS or not. Could you show the errors in the simulations or comparison between monitored and simulated debris flows?
*Our response: here, we refer to the new Appendix E and F, which shows the possible $\mu$ and $\xi$ pairs together with the corresponding z-values.*

Lines 465-469: Discussion on mineralogical composition is not very deep. The analyses of mineralogical composition just support the previous studies in Illgraben. Please consider to add discussion on the application of findings to other torrents.
*Our response: We expanded on this topic. Among the various minerals, we expect to see a control of the sheet silicates on the velocity and runout distance of the flows, mainly because clay minerals and particularly smectite-type of clays have the potential to absorb water in their crystal structure. We therefore expect that a high relative abundance of such minerals will alter the flow rheology and particularly the flows' internal friction, which is expected to impact the flow velocity. Apparently, this is not the case at the Illgraben. Yet, the relative contribution of smectite minerals, which in an Alpine setting are either the result of surface weathering or the recycling of non-metamorphosed marls and claystones, is <1% in the analyzed flow deposits. More abundant are other sheet silicates such as illites and muscovite crystals with a relative abundance between 10% and 20%. These minerals most likely formed during the low-grade Alpine metamorphism and stem from the bedrock. In addition, these sheet silicates don't have swelling properties and apparently do not impact the velocity and runout distance of the debris flows at the Illgraben. The relative high abundance of illite and muscovite crystals and the negligible content of smectite minerals points towards the importance of bedrock mass failure as the most important*

*mechanism resulting in the production of non-consolidated material in the Illgraben source area. Such a mechanism for the production of primary material was already proposed by Schlunegger et al. (2009), Berger et al. (2011) and Bennett et al. (2013) based on observations on remotely sensed datasets and is thus reconfirmed in this work. Therefore, because the debris flow material directly stems from the bedrock that experienced a low-grade metamorphosis and not from surface weathering or the recycling of non-metamorphosed claystones, the relative abundance of swelling minerals is negligible. Therefore, material which could buffer the water content in a debris flow is largely missing. This potentially explains why we don't see a correlation between the mineralogical composition of a flow and its properties such as the flow velocity and runout distance. We will discuss this in the revised version of this paper.*

**Reviewer 2:**
*Our comment: We thank reviewer 2 for the very constructive and detailed review. Many thanks!*

The manuscript uses a variety of approaches (field data, laboratory grain size analysis and modelling) to better understand debris flow dynamics in the Illgraben catchment. The study uses the well-known debris flow model, RAMMS, to simulate 13 real debris flows. These flows vary greatly in terms of velocity and flow depth, providing a good overview of different debris flow events. The authors vary the friction parameters (basal friction and viscous turbulent friction) in RAMMS and use a z-value to determine the best fit values for each debris flow event. They found that the best-fit total friction value (basal friction and viscous turbulent friction) had a significant positive correlation with field values for flow depth. Based on field data, the Froude number also provided a good indication of the debris flow dynamics and correlated well with flow volume. The study also measured the fine (<16 mm) grain size fraction of the field deposits. They found little variability in the grain size of the different flows despite stark contrasts in the flow volumes, velocities and depths measured in the field. Finally, the study also analysed the mineralogical composition of the debris flow matrix. The measured grain size distributions did not statistically significantly correlate with the field measured velocity or the input frictional parameters.

I enjoyed reading this paper and believe the authors make good use of an extensive high-resolution dataset to better understand debris flow dynamics. On the whole, the conclusions made will be of interest to the natural hazards and debris flow modelling community, particularly the observed relationships between debris flow volume, frictional parameters and the Froude number and the lack of relationship with grain size. However, I found aspects of the paper which combined field and simulated debris flow datasets confusing at times and think including specific information such as (i) which variables were simulated and which were from field datasets, (ii) how/if the z value varies between events, (iii) how well RAMMS simulated each event, is crucial to improve the clarity of the paper. I also have some general suggestions which I believe will improve the quality and readability of the manuscript. I have divided these suggestions into the sections of the manuscript and added line-by-line comments below.
*Our response: these points will be clarified in the revised manuscript. We will prepare additional tables where more information is presented on the model input and output including the related z-values. We also prepared new figures for illustration purposes, which we will include in the revised manuscript. We did not conduct further analyses, but provide such information so that the reader can better follow what we aim at documenting.*

Title/Abstract
The current title does not appear to represent the contents of the paper. I am unsure how the grain size distributions have been included in the RAMMS modelling within the paper. I inferred from the paper that the grain size did not correlate well with the input frictional parameters or debris flow dynamics but was not explicitly used in the model.
*Our response: Yes indeed, and this point has also been made by reviewer 1. We will therefore change the title to better outline what we have done.*

The abstract was well-written and set out the main findings clearly. I have noted a few typos and suggestions below.

Line 18: change "The flows" to "Previous debris flow events" to emphasise that you are simulating real events.
*Our response: Done*

Line 27: Misplaced (ii)?
*Our response: Deleted*

Lines 28 to 29: "The simulation results point to the existence of several ideal solutions" could be rephrased to note that these ideal solutions depend on the relationship between basal friction and the viscous friction parameters.
*Our response: Done, thanks for the suggestion*

Introduction

I found the hypothesis for the grain size experiment was set out clearly in the introduction and well placed within the literature. However, the structure of the introduction overall could be improved to (i) clearly define the term debris flow dynamics, (ii) to explain why it is important to understand debris flow dynamics in a natural hazard context and (iii) to better set out the hypotheses looking to be addressed.
*Our response: Done, and we have modified the introduction to better clarify the hypotheses to be tested, and we better elaborated on how we aim at testing our research question.*

The first two points can be integrated into the first paragraph.
*Our response: This has been done.*

For the main hypotheses, the authors could follow a similar format to Section 1.2 for the frictional parameters and the mineralogical analysis which is overlooked in the introduction. It would be useful to make better use of the cited literature in these sections to highlight how previous studies have related friction parameters to debris flow behaviour. This can be achieved by expanding on the references therein and adding additional references if necessary, such as the paper below.
        • Medina, V., Hürlimann, M. & Bateman, A. Application of FLATModel, a 2D finite volume code, to debris flows in the northeastern part of the Iberian Peninsula. Landslides 5, 127–142 (2008). https://doi.org/10.1007/s10346-007-0102-3
*Our response: Thanks for this reference. In the aforementioned paper, the authors tested various formula for reproducing the behaviour of debris flows. They found that the Voellmy-fluid approach reproduces the flow velocity, erosion pattern and runout distances reasonably well. We can thus use the results of this study to justify our approach, which also bases on the Voellmy-fluid rheology.*

Line 55: Only grain size is referred to here so authors could change the title accordingly.
*Our response: This has and will be done, please see response above.*

Line 67: Replace 'distinct' with 'specific'?
*Our response: We have rephrased this part to better clarify what our hypothesis will be.*

Line 78: This first sentence could be clearer. Can you expand on what is meant by the complexities outlined above? Due to the importance of grain size which is hard to measure? Or due to the variation in grain sizes transported?
*Our response: This sentence is indeed not clear, and we have removed it.*

Line 80: What type of flows does 'such flows' refer to?
*Our response: It relates to debris flows. We have specified this in the revised manuscript, and we also specify the what we mean about the behaviour.*

Line 83: Provide specific Section.
*Our response: Done*

Lines 84 to 89: I think it would be helpful for the readers to also define basal friction here as well as in the abstract.
*Our response: Done*

Section 1.3: As suggested above, this section would benefit by more reference to previous studies, previous relationships, and previous values for the friction parameters.
*Our response: This section has been rephrased to better outline the hypotheses we are going to test.*

Line 116 to 119: This sentence should be rephrased. I could not clearly see the link between grain size data and the model runs. If grain size data is used to guide model runs it should be more clearly presented throughout the manuscript. As far as I understand, the input variables for the best-fit RAMMS simulations are compared to grain size values collected in the field.
*Our response: Done, and partially shifted to the section 3.2 (methodology). We compared the field-based data of the monitored flows (velocity, volumes etc.) with the mineralogical composition and the grain size distribution and have not found a correlation. So this inference is mainly based on data from the field. We then used RAMMS to identify the relative importance of the basal and internal friction on the flow velocity and runout distance, and we found several possible solutions. Apparently, the flow properties (e.g., velocity) do depend on the flows' rheology, but this latter parameter is not dependent on the mineralogical composition of the debris flow at the Illgraben. We modified the introduction so that this chain of arguing becomes clearer.*

Lines 125: "…as crucial for the understanding of flow dynamics" This sentence refers to flow dynamics twice without stating what aspect of flow dynamics will be affected by minerology, please can you be more specific. It is also the first mention of the mineralogical component, this should also be added earlier.
*Our response: Both points have been addressed in the revised manuscript.*

Study site and setting

Line 133: Change 'This' to 'The'
*Our response: Done*

Methods

The methods section currently overlooks the approaches used to obtain the field measurements. Whilst these have been published before, the field measurements are a central component of the paper and therefore a brief overview should be included.
*Our response: We expanded on this aspect in the revised paper and outlined with more details about which data has been collected, and how data collection has been accomplished. Yet we placed this information in chapter 2 (study site and setting).*

I think the methods section would also benefit from a clear overview about how the different aspects of the paper relate. This has started to be included in lines 176 to 185 but could be clearer. The abstract was a good example of how the different results related, so a similar approach here for methods would be useful. The use of the word Because to start sentences, in my opinion, made the methods section less clear. Additional details that the authors could include to improve the repeatability of the methods are; how many model runs were conducted in RAMMS, how were the frictional values varied (at which intervals), was the number of runs the same for each debris flow event?
*Our response: The methods section has been re-structured accordingly.*

Line 179: Remove 'will' and 'those'
*Our response: We have re-written this part of the text.*

Line 181: define the input parameters that are being tested. (e.g.  $\mu$ and $\xi$)
*Our response: Done, please see also the new Appendix D, E and F.*

Line 182: define the measured debris flow properties that you are referring to. Differentiating between field and simulated values from this point would help to improve the confusion surrounding this (See comments on Results section).
*Our response: This has been done.*

Line 188: A table with the input values for the best-fit simulation for each event would be useful. How were the input event volume and peak discharge measured?
*Our response: This table is presented in the new Appendix D. The related methodology (survey in the field) is now detailed in a new section 3.1.*

Line 189: Was the drone-based DEM also 0.5 m resolution?
*Our response: The original resolution of the DEM, which we calculated from the drone images, had a resolution of 0.1 m. However, since the lidar DEM of Swisstopo has a resolution of 0.5 m, we resampled the drone-based DEM so that we could merge the Swisstopo dataset with the results of our own survey. This is not clarified in the revised text.*

Line 201: How do you define a completely developed debris flow here?
*Our response: We have modified the related section as it was not clear what we intended to say.*

Lines 210 to 214: The z value seems like a useful metric but should be included more. Do certain debris flow events have a lower z value and therefore can be simulated using RAMMS with greater certainty?
*Our response: This is presented in a new Appendix E and F.*

Lines 219: It would be useful to provide examples of basal friction values used by other studies and how the values in this study compare (this could come here or in the introduction).
*Our response: The friction coefficients are in the same range as those by other studies, which base on the Voellmy-fluid rheology. We will mention this when we present Figure 3, showing the ideal $\mu$ and $\xi$ pairs.*

A table showing how the values vary and showing all the friction values and how these alter the main outputs from RAMMS (e.g. velocity, flow depth, z value) used would also be of interest to the readers.
*Our response: We present a new table (Appendix D) where more model results (input and output together with the z-values) are displayed.*

Line 229: Please can you expand on why the levees obtain information on the dynamics of the debris flow? I would also expect the levees to be coarser than the rest of the deposit, it might be important to state this.
*Our response: This is indeed what we intend to say. We rephrased to make this point clearer. Indeed, we selected the levee deposits for three reasons. First, they can be better attributed to a specific event than other sediments. Second, these are the deposits of a debris flow that most clearly record the granulometric composition of the surge head, as our observations on video recordings have shown. Third, it is the surge head, which exerts the greatest control on the dynamics of the debris flow (Johnson et al., 2012). Accordingly, upon collecting material from levee deposits, we are likely to analyze sediments that bears the highest potential to provide information of past debris flows. Yet we acknowledge that this material is more likely coarser grained than the sediments in the tail of a debris flow (McArdell et al., 2007).*

Were all the samples collected from the same section of the deposit? This location could be added to Figure 2.
*Our response: The samples were all taken from beneath the bridge where the survey station is located. We explain this in the revised text.*

Line 231: How does this machine work? How many size fractions are the samples divided into?
*Our response: We provide more information. In particular, we processed the samples following the state-of-the-art protocol (SN670 004–2b–NA norm), which was established at the Bern University of Applied Sciences (Burgdorf). Following this protocol, the material was first dried and then sieved to a minimum particle size of 0.5 mm using a set of 7 sieves, each of which has a defined mesh size: 31.5 mm, 16 mm, 8 mm, 4 mm, 2 mm, 1 mm 0.5 mm. Subsequently, a slurry test was carried out on the material < 0.5 mm using a hydrometer. The goal of this task was to determine the particle size distribution between 0.1 and 0.001 mm. Finally, the grain size distribution between 0.5 and 0.063 mm was determined by wet sieving. During this task, we used three sieves where the mesh size was 0.25 mm, 0.125 mm and 0.063 mm.*

Line 236: Is the 1.5 to 3 kg only the fraction <16 mm? Is this just the surface? Do you have an estimate of how much of the deposit was comprised of grains >16 mm?
*Our response: We updated the related Appendix (now Appendix G), thereby presenting all information about the sieving results.*

Line 240: Change 'infer' to 'hypothesize'.
*Our response: Done.*

Line 240: Influences to influence.
*Our response: Done, thank you.*

Line 245: How do you define coarse-grained or fine-grained debris flows in this instance?
*Our response: This is now specified. We refer to the analysed spectrum of grain size.*

Results

The integration between simulated debris flow values and field measured debris flows values could be clearer. For example, lines 280 to 282 state that the main outputs from RAMMS are velocity and flow depth, which can characterize the Froude number. I am therefore confused whether the Froude numbers included in Table 1 and Figure 4 are model outputs or based on field measurements. If these are different, an additional figure comparing Froude values or an extra column in Table 1 would be helpful.
*Our response: The Froude numbers were calculated based on the flow depth and velocity values in the field. This is now clarified in the text. We did not calculate the Froude numbers from the model experiments and as such we do not compare the field- and model-based Froude numbers. We considered it more useful to compare the field- and model-based flow velocity and depth with each other. Here, we present all z-values of these comparisons in a new Appendix E, F.*

Similarly, I found the comparison between simulated and field datasets sparse. The paper would benefit from a figure directly comparing simulated velocities and flow depths to measured values as well as explicitly stating the z values. This may be better placed in the methods section or wherever the authors think is most appropriate.
*Our response: We present this information in a new table, which will be Appendix E in the revised manuscript. Instead of drafting a new figure showing this comparison, we decided to present a new Figure 5 showing the dependency of the basal and internal friction on the flow velocity and Froude numbers, as proposed by Reviewer 1.*

Lines 280 to 282: This sentence could be clearer.
*Our response: Rephrased*

Figure 4: Are the Froude numbers based on the simulated or field debris flows?
*Our response: The Froude numbers are based on measurements in the field.*

How do they compare? Could you add the z-values to this figure so that the readers can infer which pair of friction values fit best and how this varied based on the Froude number?
*Our response: This will be clarified in the revised version.*

Line 328: Are the percentages stated here relating to all events or just the fine-grained event?
*Our response: This relates to all events. We clarify this in the revised text.*

Line 333: It may be useful to note that the upper percentiles will be affected by the 16 mm cut off.
*Our response: Yes indeed, we added a cautionary note.*

Lines 341 to 351: As you deliberately tested a coarse and fine debris flow deposit, I would specifically differentiate between these in the text, even if they are not very different.
*Our response: This has been done.*

Lines 387: The end of this sentence is unclear.
*Our response: It contained a reference to a wrong figure.*

Figure 8: It would be nice if the relationships which are statistically significant are coloured or the text is shown in bold to make them easier to identify. You could also plot the individual friction values ($\mu$ and $\xi$) to see how they relate to the measured variables. Using colours in this plot could be a good way to show which variables are obtained from field data and RAMMS.
*Our response: This has been clarified. In fact, only the S-av is based on model runs. All other parameters were measured at the survey station in the field. We decided not to present the Voellmy-friction coefficients in this figure, but the values are shown in the new Appendix E.*

Figure 9: Have you tried this plot with skewness and D16, or D84/D16 I don't expect it to change but it could be interesting to test.
*Our response: Yes, but we decided not to show the results because the upper tail tends to be biased due to the reasons noted above.*

Discussion

The discussion was well written and posed some interesting findings. Additional detail could be added for the mineralogical analysis and how it has or has not provided insight into debris flow dynamics and whether it could be used to benefit future locations.

Line 420: It would be interesting if you could show evidence for this statement in Illgraben.
*Our response: This has actually been shown by the papers cited in this context. We clarify this in the revised manuscript.*

Line 452: Remove the 'and'
*Our response: Corrected*

Lines 457 to 458: Can you clarify what relationships you would expect if the GSDs were more different?
*Our response: This is now summarized in section 1.1 of the revised article, and we refer to this paragraph in the revised manuscript.*

Lines 465 to 469: It would be interesting to unpack the mineralogical composition further, perhaps more discussion about how the efficient mixing mechanism could influence debris flow dynamics and the observations made in this paper?
*Our response: This is done in the revised manuscript. Please see also suggestion by reviewer 1.*

Line 478: Could you add in brackets what you mean by dynamics here.
*Our response: Done. We consider flow velocity and also runout distance as the key parameters describing the dynamics of a debris flow.*

---

## Author Response (AR2)

Dear Editor

We have shifted the Appendix to the Supplement and modified the text accordingly. We updated the figures for a better readability.

Thank you very much for handling our manuscript.

On behalf of the authors

Fritz Schlunegger